# Adjoint sharding for very long context training of state space models

**Xingzi Xu, Amir Tavanaei, Kavosh Asadi, Karim Bouyarmane**

**Amazon**

`{xingzixu,atavanae,kavasadi,bouykari}@amazon.com`

**Reviewed on OpenReview:** `https://openreview.net/forum?id=kQCuMcEneq`

## Abstract

Despite fast progress, efficiently training large language models (LLMs) in extremely long contexts remains challenging. Existing methods fall back to training LLMs with short contexts (up to a few thousand tokens) and use inference time techniques when evaluating on very long contexts (above 1M tokens). Training on very long contexts is limited by GPU memory availability and the prohibitively long training times it requires on state-of-the-art hardware. Meanwhile, many real-life applications require training/fine-tuning with long context on specific tasks. Such applications include, for example, augmenting the context with various sources of raw reference information for extraction, summarization, or fact reconciliation tasks. We propose adjoint sharding, a novel technique that comprises sharding gradient calculation during training to reduce memory requirements by orders of magnitude, making training on very long contexts computationally tractable. At the core of our adjoint sharding algorithm lies the adjoint method, which efficiently computes gradients that are provably equivalent to the gradients computed using standard backpropagation. We also propose truncated adjoint sharding to accelerate the algorithm while maintaining performance. We provide a distributed and a parallel-computing version of adjoint sharding to speed up training and to show that adjoint sharding is compatible with these standard memory-reduction techniques. Empirical results show the proposed adjoint sharding algorithm reduces memory usage by up to $3\times$ on a large language model with 1.27B parameters on 1M context length training. This reduction in memory usage allows increasing the maximum context length of training a 1.27B parameter model from 35K tokens to above 100K tokens on a training infrastructure composed of five AWS P4 instances.

## 1 Introduction

Foundation models are a new paradigm in artificial intelligence research focused on building large, general-purpose models that adapt to different tasks OpenAI et al. (2024); Meta et al. (2024); Cai et al. (2024); Pióro et al. (2024). Extensive training on large datasets equips foundation models with broad capabilities. We then fine-tune the foundation models on smaller datasets for specific applications. Foundation models commonly employ the transformer architecture Vaswani et al. (2023). Despite the immense success, training transformer-based models requires memory growing quadratically with the context length $L$, limiting their applications on long context tasks Li et al. (2024). Researchers developed various techniques to conquer this problem, ranging from inference time context window expansion (Ding et al., 2024b;a), IO-aware algorithms (Dao et al., 2022; Dao, 2023; Shah et al., 2024), and various linearly scaling language model architectures (Gu & Dao, 2024b; Dao & Gu, 2024b; Peng et al., 2023a; Beltagy et al., 2020). On another note, distributed learning enables training large models with many GPUs, and efficient training methods like activation checkpointing, model/gradient sharding, and mixed-precision computing have further reduced the memory

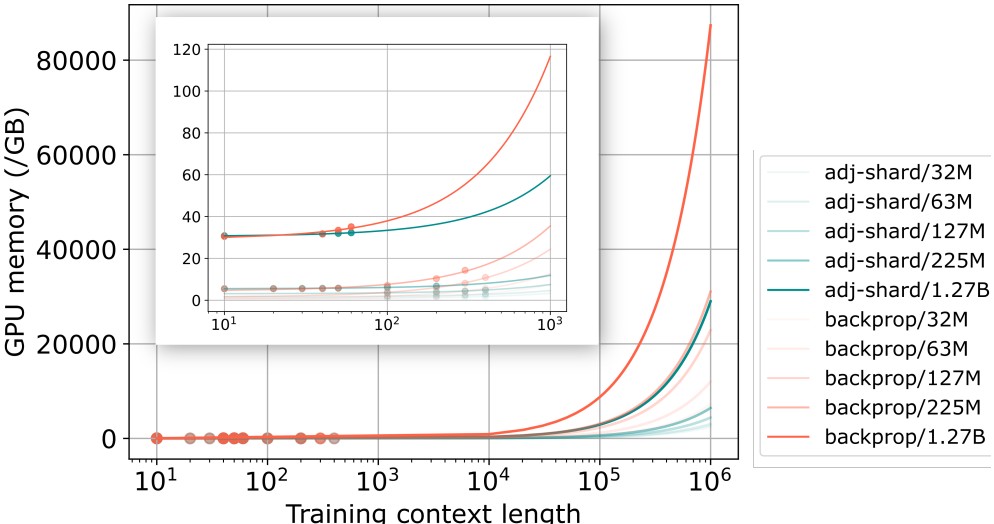

Figure 1: Compared to backpropagation (red lines), adjoint sharding (blue lines) significantly reduces memory requirements at training. Showing memory cost to train 32M, 63M, 127M, 225M, and 1.27B parameter State Space Model (SSM) with batch size 2 and Adam optimizer on one GPU.

requirement of training a large model Verbraeken et al. (2020); Zhao et al. (2023); Rajbhandari et al. (2020b); Micikevicius et al. (2018); Herrmann et al. (2019). However, current methodologies are entirely based on backpropagation and compute the gradient as a whole, inevitably requiring a memory growing rapidly with model size and context length (Damadi et al., 2023). Current sharding methods ignore the activations and only consider the model weights and optimizer states, constituting only a minor amount of the total memory cost (Sohoni et al., 2022). Activation checkpointing is among the limited techniques that consider activation values. Activation checkpointing offloads necessary intermediate states to the CPU and recomputes them on the fly, trading compute time for memory reduction (Sohoni et al., 2022; Rajbhandari et al., 2020a). The substantial time required for offloading to the CPU hinders the effectiveness of activation checkpointing. We propose adjoint sharding to disassemble the gradient computation of residual and/or recurrent-based models to achieve orders of magnitude lower memory usage during training.

**Adjoint method** The adjoint sharding method roots in the adjoint method for recurrent models (Cao et al., 2002; Johnson, 2007). Given an optimization problem of a parametric recurrent forward process, the adjoint method is concerned with the computation of the gradients regarding the process's parameters. Backpropagation saves intermediate states to calculate gradients, whereas the adjoint method relies on a backward adjoint process to compute gradients. The adjoint method is a constant-memory optimization technique for dynamical systems Chen et al. (2019); Xu et al. (2022). In this paper, we are only concerned with the adjoint method for recurrent relations.

**Vector-Jacobian product** Adjoint sharding disassembles the gradient computation of a large language model (LLM) into independent vector-Jacobian product (VJP) computations. By multiplying the Jacobian on the left with a vector, it becomes unnecessary to compute the expensive Jacobian. Modern VJPs are as fast as a forward function call of the model, and can be thousands of times faster than Jacobian computations Balestriero & Baraniuk (2021). We speed up adjoint sharding by employing the VJPs.

**Truncated adjoint sharding** Sharding the gradient computation allows us to prioritize the essential gradients and disregard the rest, resulting in faster computation. We term this novel method truncated adjoint sharding, and empirically showcase its performance.

**Distributed and parallel computation** In addition, we have developed a distributed multi-GPU variant of adjoint sharding to improve the scalability of LLM training further. We also analyze the memory cost of parallel computation of adjoint sharding, opening up directions for massive speedups.

**State-space models and residual networks** Residual networks (ResNets) are a commonly applied neural network structure. We illustrate adjoint sharding assuming a ResNet structure (He et al., 2015). State-space models (Mamba) have achieved performances on par with attention-based models while possessing a linear scaling regarding the context length $L$, a polynomial speedup compared to the $L^2$ scaling of transformers Vaswani et al. (2023); Gu & Dao (2024a).

## 2 Related works

**Linear LLMs** (De et al., 2024; Beck et al., 2024; Peng et al., 2023a) proposed LLM architectures with a linear inference time complexity. We form each linear LLM by stacking $K$ residual layers together, where each layer has a recurrent relation. However, their temporal relationships are nonlinear, which limits the application of adjoint sharding to disassemble the gradients into independent vector-Jacobian products.

**Backpropagation through time** Applying the adjoint method for recurrent models leads to backpropagation through time (BPTT) (Werbos, 1990). BPTT is a training algorithm developed for recurrent neural networks (RNNs). RNN models suffer from the exploding and vanishing gradient because of the $\prod_{j=i+1}^{t} \partial \mathbf{f}(\mathbf{x}^j, \mathbf{h}^{j-1}, \mathbf{W_h})/\partial \mathbf{h}^{j-1}$ term (Pascanu et al., 2013). SSMs provide remedies with careful parameterization of the recurrent dynamics inspired by classical SSM theory (Fu et al., 2023; Gu et al., 2021; 2022; Gupta et al., 2023; Orvieto et al., 2023; Kaul, 2020). Linear temporal relations allow efficient evaluations of the model, while preserving universal approximation capabilities (Wang & Xue, 2023). By a similar token, truncated adjoint sharding can be seen as a more general version of the truncated backpropagation through time (Jaeger, 2005; Tallec & Ollivier, 2017).

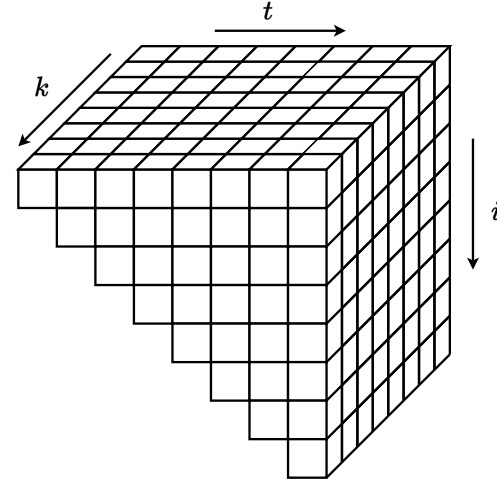

Figure 2: Adjoint sharding disassembles large models' gradient computations along the sequence dimension $t$ and the layer dimension $k$. When evaluating the gradient at time $t$, we perform $t$ vector-Jacobian products along the adjoint dimension $i$ for every layer index $k$.

**Neural ordinary differential equations** The adjoint method also applies to the optimization of continuous systems, especially the ordinary differential equations (ODEs) (Chen et al., 2019; Dupont et al., 2019). Optimizing neural ODEs with autograd requires backpropagating through numerical solvers along every step, using an unrealistic amount of memory. The adjoint method does not backpropagate through the operations of the solver and uses a constant amount of memory. However, applying the adjoint method for continuous systems requires solving a costly ODE initial value problem with dimensionality equal to the number of parameters.

**Low memory training methods** Researchers proposed various low memory training techniques to train big models in very long contexts. ZERO provides data- and model-parallel training while retaining low communication volume, while eliminating memory redundancies (Rajbhandari et al., 2020b). PyTorch FSDP streamlines model, gradient, and data parallelization (Zhao et al., 2023). Activation checkpointing discards intermediate values during the forward step, and recomputes on the fly during the training phase (Sohoni et al., 2022). CPU offloading scales large model training by offloading data and computations to the CPU, trading computing time for memory reduction (Ren et al., 2021). Ring attention leverages the blockwise computation of self-attention and feedforward to distribute long sequences across multiple devices while fully overlapping the communication of key-value blocks with blockwise attention computations, enabling very-long context training of attention-based methods (Liu et al., 2023; 2024). Mini-Sequence Transformers segment sequences but still require full gradient computation within each segment (Luo et al., 2024). Cut Your Losses optimizes vocabulary-level computations, which is orthogonal to our sequence-level memory reduction approach (Wijmans et al., 2025). StreamBP targets long sequences through a different memory-time trade-off

strategy (Luo et al., 2025). Unlike these approaches that focus on segmentation, vocabulary optimization, or memory-time trade-offs, adjoint sharding fundamentally changes the gradient computation mechanism itself by decomposing gradients into independent vector-Jacobian products that can be computed in parallel for recurrent architectures. The proposed adjoint sharding distributes state-space model computations across multiple devices and multiple multi-GPU instances (MIG) to enable very-long context training of state-space models.

**Context length extension methods** The Existing context length extension method separates into two classes. The first type is fine-tuning free methods, including Positional Interpolation (PI) (Chen et al., 2023), the NTKAware Scale ROPE (NTK) (users, 2023), and StreamingLLM (Xiao et al., 2024). The second type is fine-tuning methods, including LongChat (Li* et al., 2023), LongAlpaca (Chen et al., 2024), YaRN (Peng et al., 2023b), and LongLlama (Chen et al., 2024). Additional methods like activation beacon tune a network separate from the LLM (Zhang et al., 2024). As shown in Figure 3, fine-tuning methods achieve better performances than fine-tuning-free methods at lengths where we perform fine-tuning. However, fine-tuning methods suffer from a high computational cost and require a potentially intractable amount of GPU memory during fine-tuning.

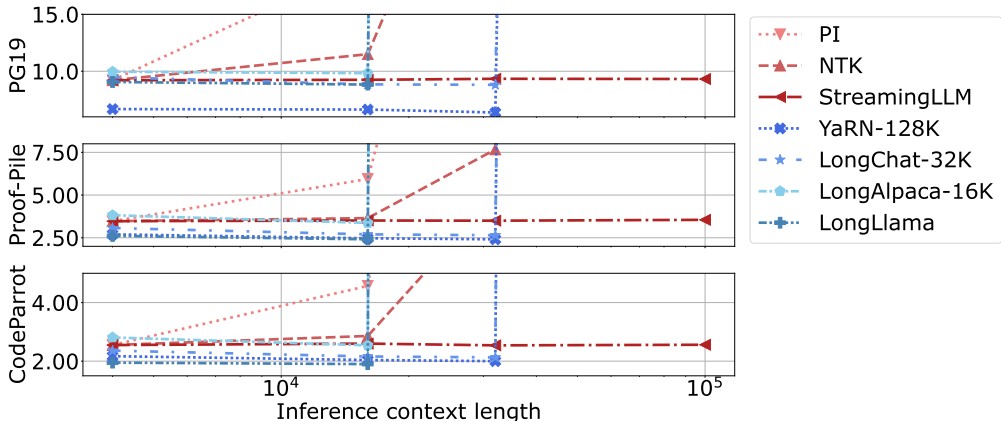

Figure 3: Lines in red and in blue are fine-tuning free and fine-tuning methods. Fine-tuning methods achieve better performances than the fine-tuning-free method but often suffer from out-of-memory issues (Chen et al., 2023; users, 2023; Xiao et al., 2024; Li* et al., 2023; Chen et al., 2024; Peng et al., 2023b; Zhang et al., 2024; Tworkowski et al., 2023). Lower values are better across all three tasks.

# 3 Background

We first give a concise introduction to the state-space models, the residual networks, and the adjoint method.

## 3.1 State-space models

While our method generally applies to all recurrent models, we illustrate the idea using state-space models (SSMs), which have shown performances at least on par with transformers at small to medium scale (Dao & Gu, 2024a). Given an input token sequence $\{\mathbf{x}_t\}_{t=1}^T$, the SSMs first calculate the corresponding matrices $\mathbf{A}^t$, $\mathbf{B}^t$, and $\mathbf{C}^t$ to evolve the dynamics as follows:

$$\mathbf{A}^t = \boldsymbol{\mathcal{A}}(\mathbf{x}^t); \; \mathbf{B}^t = \boldsymbol{\mathcal{B}}(\mathbf{x}^t); \; \mathbf{C}^t = \boldsymbol{\mathcal{C}}(\mathbf{x}^t).$$

The SSMs evolve a latent dynamics $\mathbf{h}^t$, whose initial condition $\mathbf{h}^0$ is often assumed to be zero. With $\mathbf{h}^0$ and $\mathbf{A}^t$, $\mathbf{B}^t$ defined, the dynamics evolves as:

$$\mathbf{h}^t = \mathbf{A}^t\mathbf{h}^{t-1} + \mathbf{B}^t\mathbf{x}^t.$$

The matrices $\mathbf{C}^t$ then maps the latent dynamics $\mathbf{h}^t$ back to token space as $\mathbf{y}^t = \mathbf{C}^t\mathbf{h}^t$, with $\mathbf{y}^t$ being the predicted token at $t$. For a sequence of $T$ tokens, we denote:

$$\mathbf{A} = (\mathbf{A}^1, \mathbf{A}^2, \dots, \mathbf{A}^T),\ \mathbf{B} = (\mathbf{B}^1, \mathbf{B}^2, \dots, \mathbf{B}^T),$$
$$\mathbf{C} = (\mathbf{C}^1, \mathbf{C}^2, \dots, \mathbf{C}^T),\ \mathbf{H} = (\mathbf{h}^1, \mathbf{h}^2, \dots, \mathbf{h}^T),$$
$$\mathbf{X} = (\mathbf{x}^1, \mathbf{x}^2, \dots, \mathbf{x}^T),\ \mathbf{Y} = (\mathbf{y}^1, \mathbf{y}^2, \dots, \mathbf{y}^T).$$

In the most general case, we have $\mathbf{H} \in \mathbb{R}^{T \times N}, \mathbf{A} \in \mathbb{R}^{T \times N \times N}, \mathbf{B} \in \mathbb{R}^{T \times N \times P}, \mathbf{C} \in \mathbb{R}^{T \times P \times N}, \mathbf{X} \in \mathbb{R}^{T \times P}, \mathbf{Y} \in \mathbb{R}^{T \times P}$, where $N$ is the hidden state dimension, and $P$ is the input/output dimension. We evolve the dynamics for $t = 1, \dots, T$, and assume that $\mathbf{h}^0$ is a fixed and predefined constant. The input to an SSM is $\mathbf{X}$ and $\mathbf{h}^0$, and the output is $\mathbf{Y}$. We define SSM($\cdot$) as performing the following five steps:

1. $\{\mathbf{A}^t\}_{t=1}^T = \{\mathcal{A}(\mathbf{x}^t)\}_{t=1}^T,$

2. $\{\mathbf{B}^t\}_{t=1}^T = \{\mathcal{B}(\mathbf{x}^t)\}_{t=1}^T,$

3. $\{\mathbf{C}^t\}_{t=1}^T = \{\mathcal{C}(\mathbf{x}^t)\}_{t=1}^T,$

4. $\{\mathbf{h}^t\}_{t=1}^T = \{\mathbf{A}^t\mathbf{h}^{t-1} + \mathbf{B}^t\mathbf{x}^t\}_{t=1}^T;$

5. $\{\mathbf{y}^t\}_{t=1}^T = \{\mathbf{C}^t\mathbf{h}^t\}_{t=1}^T.$

The input to the five steps is $\mathbf{X}$, and the output is $\mathbf{Y}$. We can then write SSM($\mathbf{X}$) = $\mathbf{Y}$. SSMs reduce the quadratic computational complexity with sequence length on transformers to linear, and the inference-time memory requirements from the key-value cache. SSM-based models at small to medium scales have shown performances on par with or better than transformer-based models. For instance, (Pióro et al., 2024; Anthony et al., 2024) shows that the SSM-based mixture-of-experts (MOE) model outperforms the baseline transformer-based MOE model on model sizes as big as 2400M parameters. (Waleffe et al., 2024) performed an extensive empirical study and found that while SSMs outperform transformers on various tasks, they underperform on tasks that require stellar copying, in-context learning, or long-context reasoning abilities. (Waleffe et al., 2024) also experimented with an SSM-transformer hybrid model, which outperforms transformers and is up to eight times faster when generating tokens at inference time. (Lieber et al., 2024) trained a 52B parameter model and further affirmed the hybrid model's performance.

## 3.2    Residual Networks

In practice, we have $K$ SSMs stacked together, and we have a large language head (LLH) $\Omega \in \mathbb{R}^{\mathbb{T} \times P}$, where $\mathbb{T}$ is the number of all possible tokens. To predict a token, we have $\mathbf{o}^t = \Omega\hat{\mathbf{y}}_K^t$. Define $(\mathbf{y}_K^1, \dots, \mathbf{y}_K^T) = \mathbf{Y}_K$, a ResNet computes $\mathbf{Y}_K$ as follows:

$$(\mathbf{y}_K^1, \dots, \mathbf{y}_K^T) = \mathbf{Y}_{K-1} + \text{SSM}_K(\hat{\mathbf{Y}}_{K-1})$$
$$= \mathbf{Y}_0 + \text{SSM}_1(\hat{\mathbf{Y}}_0) + \dots + \text{SSM}_K(\hat{\mathbf{Y}}_{K-1})$$
$$= \mathbf{Y}_0 + \sum_{k=1}^K \text{SSM}_k(\hat{\mathbf{Y}}_{k-1}) = \mathbf{Y}_0 + \sum_{k=1}^K \tilde{\mathbf{Y}}_k,$$

where

$$\hat{\mathbf{Y}}_k = (\hat{\mathbf{y}}_k^1, \dots, \hat{\mathbf{y}}_k^T) = (\text{Norm}(\mathbf{y}_k^1), \dots, \text{Norm}(\mathbf{y}_k^T)),$$

and $\text{SSM}_k(\hat{\mathbf{Y}}_{k-1}) = \tilde{\mathbf{Y}}_k$. Therefore, for a latent state at time $t$ we have $\mathbf{y}_K^t = \mathbf{y}_0^t + \sum_{k=1}^K \tilde{\mathbf{y}}_k^t$.

ResNet has been the foundation of numerous modern networks, including the transformers, diffusion models, segmentation models, SSMs, and more (He et al., 2016; Guo et al., 2022; Kirillov et al., 2023; Peebles & Xie, 2023). ResNet's residual structure allows for a separation between gradients of each layer by applying differentiation on summations.

### 3.3 Adjoint method

The adjoint method is concerned with optimizing $\mathbf{y}(\mathbf{h}(\boldsymbol{\theta}), \boldsymbol{\theta})$ with respect to $\boldsymbol{\theta}$, where $\mathbf{h}(\boldsymbol{\theta}) \in \mathbb{R}^P$ is the solution to $\mathbf{f}(\mathbf{h}(\boldsymbol{\theta}), \boldsymbol{\theta}) = 0$ (Cao et al., 2002). To employ gradient-based algorithms like the stochastic gradient descent (SGD) or the Adam, we compute the derivative of $\mathbf{y}$ regarding $\boldsymbol{\theta} \in \mathbb{R}^{|\boldsymbol{\theta}|}$:

$$\frac{\mathrm{d}\mathbf{y}}{\mathrm{d}\boldsymbol{\theta}} = \frac{\partial \mathbf{y}}{\partial \boldsymbol{\theta}} + \frac{\partial \mathbf{y}}{\partial \mathbf{h}} \frac{\partial \mathbf{h}}{\partial \boldsymbol{\theta}}, \tag{1}$$

with d being the total derivative, and $\partial$ being the partial derivative. The adjoint method converts computing $\mathrm{d}\mathbf{y}/\mathrm{d}\boldsymbol{\theta}$ to solving an adjoint equation. In our case, we need the adjoint method for recurrence relations, where $\mathbf{y}$ is given by $\mathbf{y} = \mathbf{y}^t \equiv \mathbf{y}(\mathbf{h}^t(\boldsymbol{\theta}), \boldsymbol{\theta})$, and $\mathbf{h}$ is given by

$$\begin{cases} \mathbf{h}^0 & = \mathbf{b}(\boldsymbol{\theta}), \\ \mathbf{h}^t & = \mathbf{f}(t, \mathbf{h}^{t-1}, \boldsymbol{\theta}). \end{cases} \tag{2}$$

We have

$$\frac{\mathrm{d}\mathbf{f}(t, \mathbf{h}^{t-1}, \boldsymbol{\theta})}{\mathrm{d}\boldsymbol{\theta}} = \frac{\partial \mathbf{f}(t, \mathbf{h}^{t-1}, \boldsymbol{\theta})}{\partial \boldsymbol{\theta}} + \frac{\partial \mathbf{f}(t, \mathbf{h}^{t-1}, \boldsymbol{\theta})}{\partial \mathbf{h}^{t-1}} \frac{\partial \mathbf{h}^{t-1}}{\partial \boldsymbol{\theta}}. \tag{3}$$

**Proposition 3.1.** *(Cao et al., 2002) When the states* $\mathbf{h}$ *are defined as Equation 2, the gradient of* $\mathbf{y}$ *with respect to* $\boldsymbol{\theta}$ *is given as:*

$$\begin{cases} \frac{\mathrm{d}\mathbf{y}^t}{\mathrm{d}\boldsymbol{\theta}} & = \frac{\partial \mathbf{y}^t}{\partial \boldsymbol{\theta}} + \boldsymbol{\lambda}^0 \mathbf{b}(\boldsymbol{\theta}) + \sum_{i=1}^t \boldsymbol{\lambda}^i \frac{\partial \mathbf{f}(i, \mathbf{h}^{i-1}, \boldsymbol{\theta})}{\partial \boldsymbol{\theta}}, \\ \boldsymbol{\lambda}^t & = \partial \mathbf{y}^t / \partial \mathbf{h}^t, \\ \boldsymbol{\lambda}^{i-1} & = \boldsymbol{\lambda}^i \left( \partial \mathbf{f}(i, \mathbf{h}^{i-1}, \boldsymbol{\theta}) / \partial \mathbf{h}^{i-1} \right). \end{cases} \tag{4}$$

*Equivalently, we have*

$$\boldsymbol{\lambda}^i = (\partial \mathbf{y}^t / \partial \mathbf{h}^t) \left( \prod_{j=t}^{i+1} \left( \partial \mathbf{f}(j, \mathbf{h}^{j-1}, \boldsymbol{\theta}) / \partial \mathbf{h}^{j-1} \right) \right)$$

*(Johnson, 2007).*

After computing adjoint states $\{\boldsymbol{\lambda}^i\}_{i=0}^t$, the computation of the elements of $\boldsymbol{\lambda}^i(\partial \mathbf{f}(i, \mathbf{h}^{i-1}, \boldsymbol{\theta}) / \partial \boldsymbol{\theta})$ are independent, allowing parallelism. This computation is a vector-Jacobian product (vjp), with $\boldsymbol{\lambda}^i$ as the vector and $\partial \mathbf{f}(i, \mathbf{h}^{i-1}, \boldsymbol{\theta}) / \partial \boldsymbol{\theta}$ as the Jacobian. vjps can be evaluated with the reverse-mode automatic differentiation and initializing the reverse phase with $\boldsymbol{\lambda}^i$ Baydin et al. (2018a). As each vjp only requires saving its corresponding computation graph, and can be disposed of after the computation, we can compute vjps in parallel on modern GPUs. We will discuss this further in subsection 4.5. Adjoint sharding aims to use the adjoint method to replace backpropagation, which solves:

$$\frac{\mathrm{d}\mathbf{y}^t}{\mathrm{d}\boldsymbol{\theta}} = \frac{\partial \mathbf{y}^t}{\partial \boldsymbol{\theta}} + \frac{\partial \mathbf{y}^t}{\partial \mathbf{h}^t} \left\{ \frac{\partial \mathbf{f}(t, \mathbf{h}^{t-1}, \boldsymbol{\theta})}{\partial \boldsymbol{\theta}} + \frac{\partial \mathbf{f}(t, \mathbf{h}^{t-1}, \boldsymbol{\theta})}{\partial \mathbf{h}^{t-1}} \left[ \frac{\partial \mathbf{f}(t-1, \mathbf{h}^{t-2}, \boldsymbol{\theta})}{\partial \boldsymbol{\theta}} + \frac{\partial \mathbf{f}(t-1, \mathbf{h}^{t-2}, \boldsymbol{\theta})}{\partial \mathbf{h}^{t-2}} \right. \right.$$
$$\left. \left. \left( \frac{\partial \mathbf{f}(t-2, \mathbf{h}^{t-3}, \boldsymbol{\theta})}{\partial \boldsymbol{\theta}} + \dots \right) \right] \right\}.$$

The backpropagation requires a sequential accumulation of the gradients, computing from the outermost layer inwards, therefore needs to save the computation graph for computations at all times $t$, and creates memory bottlenecks.

## 4 Adjoint sharding

We now introduce the adjoint sharding technique. We first illustrate the method assuming only one layer of SSM, and generalize to $K$ layers.

### 4.1 Adjoint sharding for one SSM

Large-scale neural networks usually train with the autograd framework (Baydin et al., 2018b; Paszke et al., 2019). However, this framework suffers from a high memory cost when used with networks of recurrent nature (Baydin et al., 2018b). Although activation checkpointing proves a strong tool, which discards part of the intermediate values and recomputes them later on the fly, the memory cost is still high (Herrmann et al., 2019). We employ the adjoint method for recurrence relations to reduce the memory cost further, and more importantly, to break the temporal dependencies of activations and parallelize their computations.

Define $\theta = \langle \theta_{\mathcal{A}}, \theta_{\mathcal{B}}, \theta_{\mathcal{C}} \rangle$ as $\mathcal{A}$'s, $\mathcal{B}$'s, and $\mathcal{C}$'s parameters, for loss $l^t = l(\mathbf{y}^t)$, in the context of a single-layer SSM, we prove

**Proposition 4.1.** *The gradient* $\mathrm{d}l^t/\mathrm{d}\boldsymbol{\theta}$ *is given as*

$$\frac{\mathrm{d}l^t}{\mathrm{d}\boldsymbol{\theta}} = \left[ \sum_{i=1}^{t} \mathrm{vjp}_{\mathcal{A}^i}(\frac{\mathrm{d}l^t}{\mathrm{d}\mathbf{y}^t}\boldsymbol{\lambda}^{t,i} \otimes \mathbf{h}^{i-1}) \right] \oplus \left[ \sum_{i=1}^{t} \mathrm{vjp}_{\mathcal{B}^i}(\frac{\mathrm{d}l^t}{\mathrm{d}\mathbf{y}^t}\boldsymbol{\lambda}^{t,i} \otimes \hat{\mathbf{x}}^i) \right] \oplus \mathrm{vjp}_{\mathcal{C}^t}(\frac{\mathrm{d}l^t}{\mathrm{d}\mathbf{y}^t} \otimes \mathbf{h}^t), \tag{5}$$

*where the adjoint state* $\boldsymbol{\lambda}^{t,\tau} = \mathbf{C}^t(\prod_{i=1}^{t-\tau} \mathbf{A}^{t+1-i})$, $\mathrm{vjp}_{\mathrm{Net}^i}(v) = v \cdot \mathrm{Net}_{\boldsymbol{\theta}}(\mathrm{Input}^i)$, *with* $\boldsymbol{\theta}$ *being* Net*'s parameters and* $i$ *being the index of* Input, $\otimes$ *is the vector outer product, and* $\oplus$ *is vector concatenation.*

The proof of proposition 4.1 is in section A.1. The gradient for parameters of $\mathcal{A}$, and $\mathcal{B}$ are each separated into $\{\mathrm{vjp}_{\mathcal{A}^i}(\frac{\hat{\mathrm{d}}l^t}{\mathrm{d}\mathbf{y}^t}\boldsymbol{\lambda}^{t,i} \otimes \mathbf{h}^{i-1})\}_{i=1}^{t}$, $\{\mathrm{vjp}_{\mathcal{B}^i}(\frac{\mathrm{d}l^t}{\mathrm{d}\mathbf{y}^t}\boldsymbol{\lambda}^{t,i} \otimes \hat{\mathbf{x}}^i\}_{i=1}^{t}$, and the gradient for parameters of $\mathcal{C}$ only depend on inputs at time $t$. After computing the adjoint states, these vjp computations are separated on both the network and the temporal level.

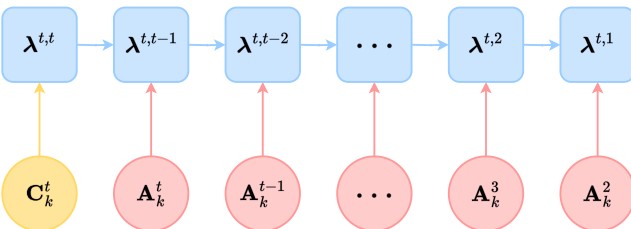

Figure 4: The adjoint states are computed sequentially backwards.

### 4.2 Adjoint sharding for multiple SSMs

We now generalize the results from subsection 4.1 to the general case of $K$ SSMs concatenated. As introduced in subsection 3.2, the outputs of each SSM layer are added to the results of the last layer and normalized before feeding into the next layer. Define the loss over all token predictions $L = \sum_{t=1}^{T} l^t$, using the residual structure we have

$$\frac{\mathrm{d}L}{\mathrm{d}\boldsymbol{\theta}} = \sum_{t=1}^{T} \frac{\mathrm{d}l^t}{\mathrm{d}\mathbf{y}_K^t} \frac{\mathrm{d}\mathbf{y}_K^t}{\mathrm{d}\boldsymbol{\theta}}$$

$$= \sum_{t=1}^{T} \frac{\mathrm{d}l^t}{\mathrm{d}\mathbf{y}_K^t} \frac{\mathrm{d}(\mathbf{y}_0^t + \sum_{k=1}^{K} \tilde{\mathbf{y}}_k^t)}{\mathrm{d}\boldsymbol{\theta}}$$

$$= \sum_{t=1}^{T} \frac{\mathrm{d}l^t}{\mathrm{d}\mathbf{y}_K^t} \sum_{k=1}^{K} \frac{\mathrm{d}\tilde{\mathbf{y}}_k^t}{\mathrm{d}\boldsymbol{\theta}}.$$

Combining with proposition 4.1, we have

**Proposition 4.2.** *The gradient of the total loss $L$ with respect to the* SSM *parameters $\boldsymbol{\theta}$ is given as*

$$\frac{\mathrm{d}L}{\mathrm{d}\boldsymbol{\theta}} = \left(\sum_{t=1}^{T}\sum_{k=1}^{K}\sum_{i=1}^{t}\mathrm{vjp}_{\mathcal{A}_k^i}(\frac{\mathrm{d}l^t}{\mathrm{d}\mathbf{y}_K^t}\boldsymbol{\lambda}_k^{t,i}\otimes\mathbf{h}_k^{i-1})\right)$$
$$\oplus\left(\sum_{t=1}^{T}\sum_{k=1}^{K}\sum_{i=1}^{t}\mathrm{vjp}_{\mathcal{B}_k^i}(\frac{\mathrm{d}l^t}{\mathrm{d}\mathbf{y}_K^t}\boldsymbol{\lambda}_k^{t,i}\otimes\hat{\mathbf{y}}_{k-1}^i)\right) \quad (6)$$
$$\oplus\left(\sum_{t=1}^{T}\sum_{k=1}^{K}\mathrm{vjp}_{\mathcal{C}_k^t}(\frac{\mathrm{d}l^t}{\mathrm{d}\mathbf{y}_K^t}\otimes\mathbf{h}_k^t)\right),$$

*where the input to* $\mathrm{vjp}_{\mathcal{C}_k^t}(\frac{\mathrm{d}l^t}{\mathrm{d}\mathbf{y}_K^t}\otimes\mathbf{h}_k^t)$, $\mathrm{vjp}_{\mathcal{A}_k^i}(\frac{\mathrm{d}l^t}{\mathrm{d}\mathbf{y}_K^t}\boldsymbol{\lambda}_k^{t,i}\otimes\mathbf{h}_k^{i-1})$, *and* $\mathrm{vjp}_{\mathcal{B}_k^i}(\frac{\mathrm{d}l^t}{\mathrm{d}\mathbf{y}_K^t}\boldsymbol{\lambda}_k^{t,i}\otimes\hat{\mathbf{y}}_{k-1}^i)$ *are computed with the $k$-th* SSM *and the* $\hat{\mathbf{y}}_{k-1}^i = \mathrm{Norm}(\mathbf{y}_{k-2}^i + \mathrm{SSM}_{k-1}(\hat{\mathbf{Y}}_{k-2})^i)$ *(the normalized output sequence of the (k-1)-th* SSM*). The adjoint state at layer $k$ is defined as* $\boldsymbol{\lambda}_k^{t,\tau} = \mathbf{C}_k^t(\prod_{i=1}^{t-\tau}\mathbf{A}_k^{t+1-i})$.

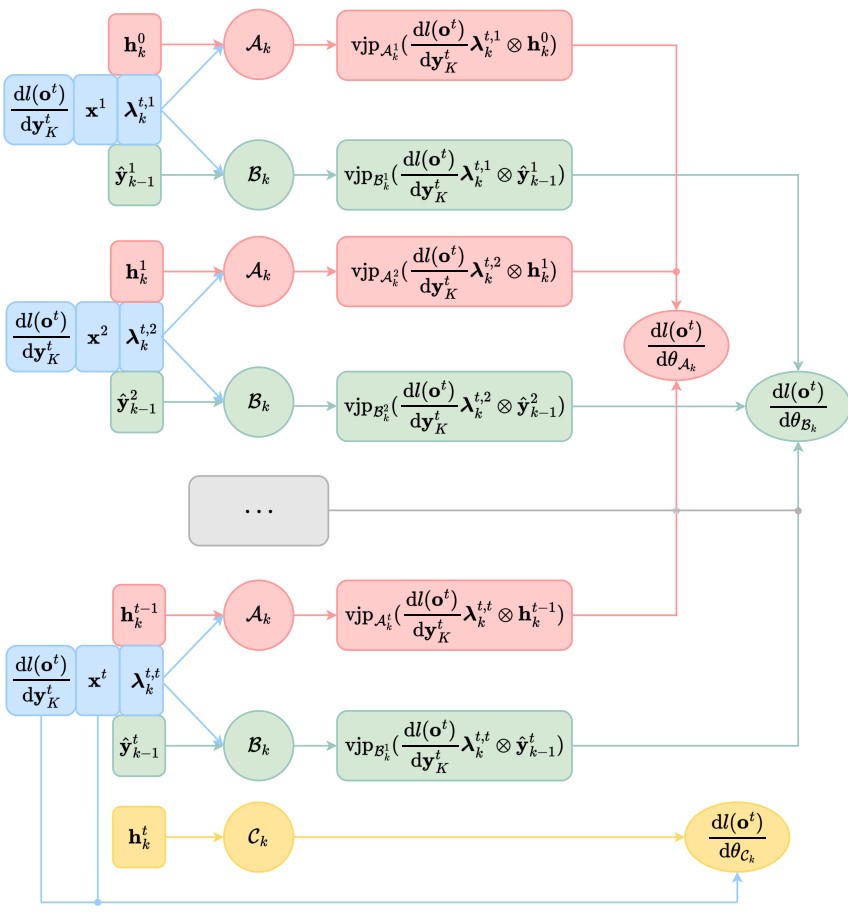

Figure 5: Computation schematic of $\mathrm{d}l^t/\mathrm{d}\boldsymbol{\theta}_{\mathcal{A}_k}$, $\mathrm{d}l^t/\mathrm{d}\boldsymbol{\theta}_{\mathcal{B}_k}$, and $\mathrm{d}l^t/\mathrm{d}\boldsymbol{\theta}_{\mathcal{C}_k}$.

We provide the proof of proposition 4.2 in section A.2. Define $\boldsymbol{\Lambda}_k^t = \{\boldsymbol{\lambda}_k^{t,\tau}\}_{\tau=1}^t$, proposition 4.2 shows that the gradients of each network's parameters computed with each token only correlate through the adjoint states $\{\boldsymbol{\Lambda}_k^t\}_{k,t=1,1}^{K,T}$. We can efficiently compute the adjoint states after a forward pass. We can also compute the adjoint states on the fly in the gradient computation phase, as it only depends on $\mathbf{C}_k^t$ and $\mathbf{A}_k^t$ and has no dependencies on the network Jacobians regarding the network parameters. The adjoint sharding method breaks down the backpropagation computation from layer-wise and token-wise into foundational vjp computations that do not have any dependencies on each other.

We show a schematic of the computations to $\mathrm{d}l^t/\mathrm{d}\boldsymbol{\theta}_{\mathcal{A}_k}$, $\mathrm{d}l^t/\mathrm{d}\boldsymbol{\theta}_{\mathcal{B}_k}$, and $\mathrm{d}l^t/\mathrm{d}\boldsymbol{\theta}_{\mathcal{C}_k}$ in Figure 5 and a schematic for computing the adjoint states in Figure 4.

## 4.3 Truncated adjoint sharding

One limitation of adjoint sharding is that the number of vjps performed increases polynomially with the number of tokens $T$. In particular, adjoint sharding computes the vjp for $\mathcal{A}_k$ and $\mathcal{B}_k$ $(1+T)T/2$ times, and for $\mathcal{C}_k$ $T$ times. When training large networks with many layers and a long context length $T$, applying adjoint sharding becomes computationally expensive. We propose truncated adjoint sharding, with which we argue that we can get similar results by computing a linearly growing number of vjps, and empirically showcase its performance.

Attention mechanisms have suffered from the $\mathcal{O}(T^2)$ complexities arising from the self-attention structure (Vaswani et al., 2023). To enable training with longer context lengths, global-local attention has been proposed, where we divide the contexts into sections, and compute the attention between sections rather than tokens (Yang et al., 2021). (Tallec & Ollivier, 2017) proposed truncated backpropagation through time (T-BPTT) to avoid gradient explosion/vanishing when training with very long contexts by only counting a fixed number of state transitions. Here, inspired by global-local attention and T-BPTT, instead of computing the full gradient given in Equation 11, we propose to train the SSMs to depend on up to $\bar{T}$ states:

$$\frac{\mathrm{d}L}{\mathrm{d}\boldsymbol{\theta}} = \left(\sum_{t=1}^{T}\sum_{k=1}^{K}\mathrm{vjp}_{\mathcal{C}_k^t}\left(\frac{\mathrm{d}l^t}{\mathrm{d}\mathbf{y}_K^t}\otimes\mathbf{h}_k^t\right)\right) \oplus \left[\sum_{t=1}^{\bar{T}}\sum_{k=1}^{K}\sum_{i=1}^{t}\mathrm{vjp}_{\mathcal{A}_k^i}\left(\frac{\mathrm{d}l^t}{\mathrm{d}\mathbf{y}_K^t}\boldsymbol{\lambda}_k^{t,i}\otimes\mathbf{h}_k^{i-1}\right)\right.$$
$$+ \sum_{t=\bar{T}+1}^{T}\sum_{k=1}^{K}\sum_{i=t+1-\bar{T}}^{t}\mathrm{vjp}_{\mathcal{A}_k^i}\left(\frac{\mathrm{d}l^t}{\mathrm{d}\mathbf{y}_K^t}\boldsymbol{\lambda}_k^{t,i}\otimes\mathbf{h}_k^{i-1}\right)\right] \oplus \left[\sum_{t=1}^{\bar{T}}\sum_{k=1}^{K}\sum_{i=1}^{t}\mathrm{vjp}_{\mathcal{B}_k^i}\left(\frac{\mathrm{d}l^t}{\mathrm{d}\mathbf{y}_K^t}\boldsymbol{\lambda}_k^{t,i}\otimes\hat{\mathbf{y}}_{k-1}^i\right)\right. \quad (7)$$
$$\left. + \sum_{t=\bar{T}+1}^{T}\sum_{k=1}^{K}\sum_{i=t+1-\bar{T}}^{t}\mathrm{vjp}_{\mathcal{B}_k^i}\left(\frac{\mathrm{d}l^t}{\mathrm{d}\mathbf{y}_K^t}\boldsymbol{\lambda}_k^{t,i}\otimes\hat{\mathbf{y}}_{k-1}^i\right)\right]$$

As shown in Equation 7 above, we perform the same computations for $t = 1, \ldots, \bar{T}$ as before, and only perform the vjps back to the last $\bar{T}$ states for $t > \bar{T}$. With truncated adjoint sharding, we perform $\bar{T}T + \bar{T}(\bar{T}-1)/2$ vjps, which grows linearly with $T$. We show the number of vjps performed with and without truncated adjoint sharding in Figure 6. When $\bar{T} = 2000$, truncated adjoint sharding reduces 64% of the vjps when training with a context length of 10K. The essence of the truncated adjoint sharding method is that we only explicitly count gradients related to the last $\bar{T}$ states. As each state depends on its prior state, states still implicitly depend on all their prior states. The choice of truncation parameter $\bar{T}$ involves a trade-off between computational efficiency and gradient fidelity. In practice, we find choosing $\bar{T} = \sqrt{T}$ provides a good balance between memory savings and gradient fidelity, as it keeps computation time in check while preserving sufficient gradient information. The optimal choice depends on the specific model architecture and sequence characteristics. We leave the investigation of $\bar{T}$'s impact on performances for future work.

We provide the algorithm for evaluating adjoint states for token index $t$ and ResNet index $k$ with truncated adjoint sharding $\bar{T}$ in algorithm 1, and the algorithm for evaluating the vjps for token index $t$ and ResNet index $k$ with truncated adjoint sharding $\bar{T}$ in algorithm 2.

## 4.4 Distributed training

We now discuss how to distribute the storage and compute of the adjoint sharding method, assuming that we have $\Upsilon$ GPUs. Given the networks $\{\mathcal{A}_k, \mathcal{B}_k, \mathcal{C}_k\}_{k=1}^{K}$, initial tokens $\{\hat{\mathbf{y}}_0^t\}_{t=1}^{T} = \{\mathrm{Norm}(\mathbf{x}^t)\}_{t=1}^{T}$, and initial conditions $\{\mathbf{h}_k^0\}_{k=1}^{K}$ (usually set to $\mathbf{0}$), we can call algorithm 3 to get all necessary vectors for computing the gradient with adjoint sharding. As shown in algorithm 2, to compute the vjps' for token index $t$ and ResNet index $k$, we only need $t, k, \mathrm{d}l(\mathbf{o}^t)/\mathrm{d}\mathbf{y}_K^t, \{\mathbf{h}_k^i\}_{i=0}^{t}, \mathbf{C}_k^t, \{\hat{\mathbf{y}}_{k-1}^i\}_{i=1}^{t}, \{\mathbf{A}_k^i\}_{i=2}^{t}$. To compute all the gradients for layer $k$, we only need $\mathbf{A}$, $\mathbf{h}$, and $\mathbf{C}$ from the $k$-th layer, and $\hat{\mathbf{y}}$ from the $k-1$-th layer. Therefore, we can divide the $K$ layers into $\Upsilon$ pieces, as shown in the appendix B. As the computations are fully independent

---

**Algorithm 1** Evaluating adjoint states for token index $t$ and ResNet index $k$ with truncated adjoint sharding $\bar{T}$

---

1: **Inputs:** $t$, $k$, $\bar{T}$, $\mathbf{C}_k^t$, $\{\mathbf{A}_k^i\}_{i=t+2-\bar{T}}^t$
2: Initialize adjoint state $\boldsymbol{\lambda}_k^{t,t} = \mathbf{C}_k^t$
3: Compute: intermediate values:
4: $\boldsymbol{\zeta}^{\bar{T}} = (\mathbf{A}_k^t \mathbf{A}_k^{t-1} \dots \mathbf{A}_k^{t+2-\bar{T}},$
   $\mathbf{A}_k^t \mathbf{A}_k^{t-1} \dots \mathbf{A}_k^{t+3-\bar{T}}, \dots, \mathbf{A}_k^t \mathbf{A}_k^{t-1}, \mathbf{A}_k^t, \mathbb{I}).$
5: Compute: adjoint states
6: $\bar{\boldsymbol{\Lambda}}_k^{\bar{T}} = (\boldsymbol{\lambda}_k^{t,t+1-\bar{T}}, \boldsymbol{\lambda}_k^{t,t+2-\bar{T}}, \dots, \boldsymbol{\lambda}_k^{t,t}) = \mathbf{C}_k^t \boldsymbol{\zeta}^{\bar{T}}.$
7: **Return:** $\bar{\boldsymbol{\Lambda}}_k^{\bar{T}}$.

---

**Algorithm 2** Evaluating the vjp's for token index $t$ and ResNet index $k$ with truncated adjoint sharding $\bar{T}$

---

1: **Inputs:** $t$, $k$, $\bar{T}$, $\frac{\mathrm{d}l(\mathbf{o}^t)}{\mathrm{d}\mathbf{y}_K^t}$, $\{\mathbf{h}_k^i\}_{i=t-\bar{T}}^t$, $\mathbf{C}_k^t$, $\{\mathbf{y}_{k-1}^i\}_{i=t+1-\bar{T}}^t$, $\{\mathbf{A}_k^i\}_{i=t+2-\bar{T}}^t$
2: **Call alg. 1 to compute** $\{\boldsymbol{\lambda}_k^{t,i}\}_{i=t+1-\bar{T}}^t$
3: **Compute:** $\frac{\mathrm{d}l(\mathbf{o}^t)}{\mathrm{d}\mathbf{y}_K^t} \otimes \mathbf{h}_k^t$, $\{\frac{\mathrm{d}l(\mathbf{o}^t)}{\mathrm{d}\mathbf{y}_K^t}\boldsymbol{\lambda}_k^{t,i} \otimes \mathbf{h}_k^{i-1}\}_{i=t+1-\bar{T}}^t$, $\{\frac{\mathrm{d}l(\mathbf{o}^t)}{\mathrm{d}\mathbf{y}_K^t}\boldsymbol{\lambda}_k^{t,i} \otimes \hat{\mathbf{y}}_{k-1}^i\}_{i=t+1-\bar{T}}^t$
4: **Compute:** $\left(\mathrm{vjp}_{\mathbf{C}_k^t}(\frac{\mathrm{d}l(\mathbf{o}^t)}{\mathrm{d}\mathbf{y}_K^t} \otimes \mathbf{h}_k^t), \sum_{i=t+1-\bar{T}}^t \mathrm{vjp}_{\mathbf{A}_k^i}(\frac{\mathrm{d}l(\mathbf{o}^t)}{\mathrm{d}\mathbf{y}_K^t}\boldsymbol{\lambda}_k^{t,i} \otimes \mathbf{h}_k^{i-1}), \sum_{i=t+1-\bar{T}}^t \mathrm{vjp}_{\mathbf{B}_k^i}(\frac{\mathrm{d}l(\mathbf{o}^t)}{\mathrm{d}\mathbf{y}_K^t}\boldsymbol{\lambda}_k^{t,i} \otimes \hat{\mathbf{y}}_{k-1}^i)\right)$
5: **Return:** $\left(\mathrm{vjp}_{\mathbf{C}_k^t}(\frac{\mathrm{d}l(\mathbf{o}^t)}{\mathrm{d}\mathbf{y}_K^t} \otimes \mathbf{h}_k^t), \sum_{i=t+1-\bar{T}}^t \mathrm{vjp}_{\mathbf{A}_k^i}(\frac{\mathrm{d}l(\mathbf{o}^t)}{\mathrm{d}\mathbf{y}_K^t}\boldsymbol{\lambda}_k^{t,i} \otimes \mathbf{h}_k^{i-1}), \sum_{i=t+1-\bar{T}}^t \mathrm{vjp}_{\mathbf{B}_k^i}(\frac{\mathrm{d}l(\mathbf{o}^t)}{\mathrm{d}\mathbf{y}_K^t}\boldsymbol{\lambda}_k^{t,i} \otimes \hat{\mathbf{y}}_{k-1}^i)\right)$

---

**Algorithm 3** Forward step in evaluation mode on a distributed system

---

1: **Inputs:** $\{\hat{\mathbf{y}}_0^t\}_{t=1}^T$, $\{\mathbf{h}_k^0\}_{k=1}^K$, $\{\mathcal{A}_k, \mathcal{B}_k, \mathcal{C}_k\}_{k=1}^K$, $\Omega$
2: On devices $v = 1, \dots, \Upsilon$, in parallel **do**
3: **for** SSM model index $k = (v-1)(K//\Upsilon) + 1, \dots, v(K//\Upsilon)$ **do**
4:     **for** Time step index $t = 1, \dots, T$ **do**
5:         Compute: $\mathbf{A}_k^t = \mathcal{A}_k(\hat{\mathbf{y}}_{k-1}^t)$; $\mathbf{B}_k^t = \mathcal{B}_k(\hat{\mathbf{y}}_{k-1}^t)$; $\mathbf{C}_k^t = \mathcal{C}_k(\hat{\mathbf{y}}_{k-1}^t)$; $\mathbf{h}_k^t = \mathbf{A}_k^t \mathbf{h}_k^{t-1} + \mathbf{B}_k^t \hat{\mathbf{y}}_{k-1}^t$; $\mathbf{y}_k^t = \mathbf{C}_k^t \mathbf{h}_k^t$.
6:         Compute: $\mathbf{y}_k^t = \mathbf{y}_{k-1}^t + \hat{\mathbf{y}}_k^t$.
7:         Compute: $\hat{\mathbf{y}}_k^t = \mathrm{Norm}(\mathbf{y}_k^t)$.
8:     **end for**
9: **end for**
10: Store: $\quad \{\mathbf{h}_k^t\}_{(t,k)=(1,(v-1)(K//\Upsilon)+1)}^{T,v(K//\Upsilon)}, \quad \{\mathbf{C}_k^t\}_{(t,k)=(1,(v-1)(K//\Upsilon)+1)}^{T,v(K//\Upsilon)}, \quad \{\hat{\mathbf{y}}_k^t\}_{(t,k)=(1,(v-1)(K//\Upsilon))}^{T,v(K//\Upsilon)-1},$
   $\{\mathbf{A}_k^t\}_{(t,k)=(2,(v-1)(K//\Upsilon)+1)}^{T,v(K//\Upsilon)}$ on device $v$.
11: Pass: $\{\mathbf{y}_{v(K//\Upsilon)-1}^t\}_{t=1}^T$, $\{\hat{\mathbf{y}}_{v(K//\Upsilon)-1}^t\}_{t=1}^T$ to device $v+1$
12: **for** Time step index $t = 1, \dots, T$ **do**
13:     Compute: $\{\mathbf{o}^t = \Omega \mathbf{y}_K^t\}_{t=1}^T$, $\{l(\mathbf{o}^t)\}$, $\{\frac{\mathrm{d}l(\mathbf{o}^t)}{\mathrm{d}\mathbf{y}_K^t}\}_{t=1}^T$.
14: **end for**
15: Store: $\{\frac{\mathrm{d}l(\mathbf{o}^t)}{\mathrm{d}\mathbf{y}_K^t}\}_{t=1}^T$ on all $\Upsilon$ devices.

---

and we compute the gradients using only data on local devices, we additionally distribute the model and the gradients, as shown in Table 6, where $\boldsymbol{\theta}_k$ represents the parameters of $\mathcal{A}_k$, $\mathcal{B}_k$, and $\mathcal{C}_k$, and Gradient$_k$ represents the optimizer states for $\boldsymbol{\theta}_k$. The complete training streamline is provided in algorithm 4. We distribute the activations, computations, gradients, and optimization states across $\Upsilon$ devices. While the forward evaluation pass results across different devices, as shown in algorithm 3, the computation of gradients is parallel across the $\Upsilon$ devices. This parallelization will speed up the training as the gradient computation takes most of the computation budget. We will also get a memory per GPU close to Mem/$\Upsilon$, with Mem being the memory cost if we only have a single GPU. If we have $\Upsilon > K$ devices, we can further speed up the forward evaluation by first evaluating $\mathcal{A}$, $\mathcal{B}$, $\mathcal{C}$ in parallel, and then adding them together on the distributed devices.

**Algorithm 4** Evaluating $\frac{\mathrm{d}L}{\mathrm{d}\boldsymbol{\theta}}$ with truncated adjoint sharding $\bar{T}$ on $\Upsilon$ devices

1: **Inputs:** $\{\mathbf{y}_0^t\}_{t=1}^T$, $\{\mathbf{h}_k^0\}_{k=1}^K$, $\{\mathcal{A}_k, \mathcal{B}_k, \mathcal{C}_k\}_{k=1}^K$, $\Omega$, $\bar{T}$, $\Upsilon$
2: Call alg. 3 for $\{\mathbf{A}_k^t, \mathbf{C}_k^t, \mathbf{h}_k^t, \hat{\mathbf{y}}_k^t\}_{(t,k)=(1,1)}^{(T,K)}$, $\{\frac{\mathrm{d}l(\mathbf{o}^t)}{\mathrm{d}\mathbf{y}_K^t}\}_{t=1}^T$ and saved on each GPU device.
3: On each device $v$, in parallel **do**
4: Initialize gradient $\frac{\mathrm{d}L}{\mathrm{d}\boldsymbol{\theta}}$
5: **for** Time step index $t = 1, \ldots, \bar{T}$, layer index $k = (v-1)(K//\Upsilon) + 1, \ldots, v(K//\Upsilon)$ **do**
6: $\quad$ Call alg. 2 for $\Xi = \left[ \text{vjp}_{\mathbf{C}_k^t}(\frac{\mathrm{d}l(\mathbf{o}^t)}{\mathrm{d}\mathbf{y}_K^t} \otimes \mathbf{h}_k^t), \sum_{i=1}^t \text{vjp}_{\mathbf{A}_k^i}(\frac{\mathrm{d}l(\mathbf{o}^t)}{\mathrm{d}\mathbf{y}_K^t} \boldsymbol{\lambda}_k^{t,i} \otimes \mathbf{h}_k^{i-1}), \sum_{i=1}^t \text{vjp}_{\mathbf{B}_k^i}(\frac{\mathrm{d}l(\mathbf{o}^t)}{\mathrm{d}\mathbf{y}_K^t} \boldsymbol{\lambda}_k^{t,i} \otimes \hat{\mathbf{y}}_{k-1}^i) \right]$
7: $\quad$ Compute: $\frac{\mathrm{d}L}{\mathrm{d}\boldsymbol{\theta}} += \Xi$
8: **end for**
9: **for** Time step index $t = \bar{T}+1, \ldots, T$, layer index $k = (v-1)(K//\Upsilon) + 1, \ldots, v(K//\Upsilon)$ **do**
10: $\quad$ Call alg. 2 for $\Xi = \left( \text{vjp}_{\mathbf{C}_k^t}(\frac{\mathrm{d}l(\mathbf{o}^t)}{\mathrm{d}\mathbf{y}_K^t} \otimes \mathbf{h}_k^t), \sum_{i=t+1-\bar{T}}^t \text{vjp}_{\mathbf{A}_k^i}(\frac{\mathrm{d}l(\mathbf{o}^t)}{\mathrm{d}\mathbf{y}_K^t} \boldsymbol{\lambda}_k^{t,i} \otimes \mathbf{h}_k^{i-1}), \sum_{i=t+1-\bar{T}}^t \text{vjp}_{\mathbf{B}_k^i}(\frac{\mathrm{d}l(\mathbf{o}^t)}{\mathrm{d}\mathbf{y}_K^t} \boldsymbol{\lambda}_k^{t,i} \otimes \hat{\mathbf{y}}_{k-1}^i) \right)$
11: $\quad$ Compute: $\frac{\mathrm{d}L}{\mathrm{d}\boldsymbol{\theta}} += \Xi$
12: **end for**
13: **Return:** $\frac{\mathrm{d}L}{\mathrm{d}\boldsymbol{\theta}}$

## 4.5 Parallel computing

Adjoint sharding converts the sequential process of backpropagation gradient computation into individual independent vjps, allowing for parallel computation. We analyze the time and memory cost of $\text{vjp}_{\mathcal{A}_k^i}((\mathrm{d}l^t/\mathrm{d}\mathbf{y}_K^t)\boldsymbol{\lambda}_k^{t,i} \otimes \mathbf{h}_k^{i-1})$, $\text{vjp}_{\mathcal{B}_k^i}((\mathrm{d}l^t/\mathrm{d}\mathbf{y}_K^t)\boldsymbol{\lambda}_k^{t,i} \otimes \hat{\mathbf{y}}_{k-1}^i)$, and $\text{vjp}_{\mathcal{C}_k^t}((\mathrm{d}l^t/\mathrm{d}\mathbf{y}_K^t) \otimes \mathbf{h}_k^t)$. vjp has a similar time complexity as a forward pass, and a memory complexity of $\text{bs}(|\boldsymbol{\theta}| + \mathbb{O}) + |\boldsymbol{\theta}|$, where bs is the batch size, $\mathbb{O}$ is the number of elements in the network output, and $|\boldsymbol{\theta}|$ is the number of parameters (Novak et al., 2022). We provide the memory and FLOPs required to compute the vjps in Table 1 (NVIDIA, 2024).

We analyze training with a dataset containing contexts of lengths $T$, with $\Upsilon$ NVIDIA H100 GPUs, and performing computations in FP16. We use a selective diagonal SSM with $K$ layers, and each $\mathcal{A}_k$, $\mathcal{B}_k$, and $\mathcal{C}_k$ network is a single-layer multi-layer perceptron (MLP). For each data point $\{\mathbf{x}^t\}_{t=1}^T$, we store $\{\mathbf{A}_k^t, \mathbf{C}_k^t, \mathbf{h}_k^t, \mathbf{y}_k^t\}_{(t,k)=(1,1)}^{(T,K)}$ and $\{\mathrm{d}l(\mathbf{o}^t)/\mathrm{d}\mathbf{y}_K^t\}_{t=1}^T$, which is $TK(2N+P) + TP$ FP16 numbers. We also save $\boldsymbol{\theta}_\mathcal{A}$, $\boldsymbol{\theta}_\mathcal{B}$, and $\boldsymbol{\theta}_\mathcal{C}$, each taking $PN + N$ FP16 numbers. We need to store $T(2NK + PK + P) + 3N(P+1)$ FP16 numbers before computing the vjp. As computing all adjoint state sequences takes up to $N(2P+1)(1+T)T/2$

| | | vjp$_\mathcal{A}$ | vjp$_\mathcal{B}$ | vjp$_\mathcal{C}$ |
|---|---|---|---|---|
| Unstructured SSM | Memory | $\text{bs}(N^2 + |\boldsymbol{\theta}_\mathcal{A}|^*) + |\boldsymbol{\theta}_\mathcal{A}|$ | $\text{bs}(NP + |\boldsymbol{\theta}_\mathcal{B}|^*) + |\boldsymbol{\theta}_\mathcal{B}|$ | $\text{bs}(NP + |\boldsymbol{\theta}_\mathcal{C}|^*) + |\boldsymbol{\theta}_\mathcal{C}|$ |
| | FLOPs | $\text{bs}(N^2(2P+1))$ | $\text{bs}(NP(2P+1))$ | $\text{bs}(NP \times (2P+1))$ |
| Diagonal SSM | Memory | $\text{bs}(N + |\boldsymbol{\theta}_\mathcal{A}|^*) + |\boldsymbol{\theta}_\mathcal{A}|$ | $\text{bs}(N + |\boldsymbol{\theta}_\mathcal{B}|^*) + |\boldsymbol{\theta}_\mathcal{B}|$ | $\text{bs}(N + |\boldsymbol{\theta}_\mathcal{C}|^*) + |\boldsymbol{\theta}_\mathcal{C}|$ |
| | FLOPs | $\text{bs}(N(2P+1))$ | $\text{bs}(N(2P+1))$ | $\text{bs}(N(2P+1))$ |
| Scalar SSM | Memory | $\text{bs}(1 + |\boldsymbol{\theta}_\mathcal{A}|^*) + |\boldsymbol{\theta}_\mathcal{A}|$ | $\text{bs}(N + |\boldsymbol{\theta}_\mathcal{B}|^*) + |\boldsymbol{\theta}_\mathcal{B}|$ | $\text{bs}(N + |\boldsymbol{\theta}_\mathcal{C}|^*) + |\boldsymbol{\theta}_\mathcal{C}|$ |
| | FLOPs | $\text{bs}(2P+1)$ | $\text{bs}((N(2P+1))$ | $\text{bs}(N(2P+1))$ |

Table 1: Memory and FLOPs required to compute the vjps. $|\boldsymbol{\theta}_\mathcal{A}|^*$, $|\boldsymbol{\theta}_\mathcal{B}|^*$, and $|\boldsymbol{\theta}_\mathcal{C}|^*$ represents the number of elements of the biggest parameter vector of $\mathcal{A}$, $\mathcal{B}$, and $\mathcal{C}$.

FLOPs, it takes $NP(1+T)/T$ FLOPs on average for each adjoint state. For $T$ large enough, $(1+T)/T \approx 1$, we approximate the average FLOPs for each adjoint state with $NP$. Each vjp then takes $\text{bs}(7NP + 3N)$ FLOPs of computation.

When computing with a selective diagonal SSM with $P = 128$, $N = 225$, and bs $= 8$, while storing and performing computations in FP16, computing vjp$_\mathcal{A}$, vjp$_\mathcal{B}$, and vjp$_\mathcal{C}$ each takes around 0.6MB memory and 1798144 FLOPs. We characterize the capacity of a modern GPU with FLOPs/sec, which measures the computation speed; GPU memory bandwidth, which is the rate at which a GPU can move data between its memory and processing cores; GPU Memory, which is the amount of data a GPU can hold; and number of Multi-Instance GPU (MIG) instances, which is the number of fully isolated GPU instances with its own high-bandwidth memory, cache, and compute cores a GPU can host. An NVIDIA H100 Tensor Core GPU has a

GPU memory bandwidth 3.35TB/s and performs $1,979$ tera FP16 FLOPS per second. Therefore, the memory bandwidth allows computing $(3.35\text{TB/s})/0.6\text{MB} = 5.58 \times 10E6$ batches of vjps per second, and the computing speed allows computing $(1979\text{tera/s})/1798144 = 3.76 \times 1.1E9$ batches of vjps per second. At the same time, since the H100 GPU has 80GB memory, it can hold up to $80\text{GB}/(0.6\text{MB/vjp}) = 133$ batches of vjps at the same time if we do not consider any memory overhead. As each H100 GPU can hold up to $7$ instances in parallel, we perform the adjoint sharding algorithm with $7\Upsilon$ instances, offering as much as a 56x speedup on one AWS P4 instance (8 H100 GPUs). We can not achieve such speedup for backpropagation because of its sequential nature.

**Limitation** The adjoint sharding method provides an alternative method of computing gradients to backpropagation. While we analytically proved that the gradients computed from adjoint sharding are equal to those from backpropagation, adjoint sharding suffers from a polynomial time complexity regarding the training context length when computing equivalent gradients. We provided the truncated adjoint sharding as a linear time complexity alternative, and leave the analysis of its convergence and further improvements on it for future work. We also provided a distributed and parallel computing algorithm for performing adjoint sharding. However, the overhead of naïve implementations of such an algorithm with multithreading or multiprocessing overweights the speedups when the training context length is small. We leave the efficient implementation of the parallel algorithm on a CUDA kernel for future work. Our work focuses specifically on State Space Models (SSMs) like Mamba, which

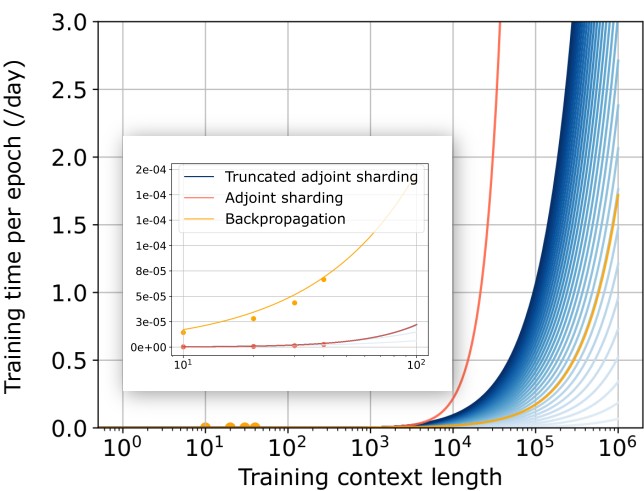

Figure 6: Training time (/day) per epoch comparison for adjoint sharding, truncated adjoint sharding, and backpropagation with different context lengths. Assumed a 100-layer SSM-ResNet model, a 280x acceleration for adjoint sharding from parallel computing (achievable with five Amazon P4 instances), and $\bar{T}$ from 15 to 2500.

have fundamentally different computational structures than transformers. The adjoint sharding method we developed leverages the recurrent nature of SSMs, where the sequential dependencies allow for efficient VJP decomposition. Transformers, with their attention mechanisms and different architectures, would require a separate analysis to determine how to apply adjoint methods. Unlike SSMs' Markovian recurrent structure that enables clean temporal decomposition, transformers' self-attention creates all-to-all dependencies that break the sequential dependency structure the adjoint method relies on. This architectural constraint represents an important direction for future work, as extending memory-efficient training techniques to Transformer architectures would significantly broaden the impact of this approach.

**Conclusion** We introduced adjoint sharding, a distributed and parallel computing algorithm, to facilitate the training of LLMs on very long contexts. Unlike the sequential backpropagation, the adjoint sharding computes gradients of each LLM layer against each token independently through vector-Jacobian product, allowing for parallel computation. We propose truncated adjoint sharding to focus on essential gradients to avoid the limitation of vjps increasing polynomially regarding context length. Our approach provides an algorithmic alternative to existing memory reduction techniques such as activation checkpointing, model sharding, and vocabulary optimization methods. While these techniques address different aspects of the memory bottleneck, adjoint sharding specifically targets the gradient computation mechanism in recurrent architectures. The method is currently applicable to State Space Models, where the sequential recurrent structure enables clean VJP decomposition. We analyzed the memory and FLOP cost of each computation block in adjoint sharding and proposed a method to accelerate it through parallel computing. Empirical results suggest orders of magnitude of memory reduction in training while maintaining the same training results as backpropagation. By enabling training on contexts previously intractable due to memory constraints, adjoint sharding opens new possibilities for fine-tuning SSM-based models on very long sequence tasks.

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

# A Proof

## A.1 Proof for proposition 4.1

*Proof.* Define $\partial\tilde{\mathbf{y}}/\partial\mathbf{h}^t = \tilde{\mathbf{y}}_{\mathbf{h}^t}^t$, $\partial\tilde{\mathbf{h}}^t/\partial\mathbf{h}^{t-1} = \tilde{\mathbf{h}}_{\mathbf{h}^{t-1}}^t$, and $\partial\tilde{\mathbf{y}}/\partial\boldsymbol{\theta} = \tilde{\mathbf{y}}_{\boldsymbol{\theta}}^t$, $\partial\tilde{\mathbf{h}}^t/\partial\boldsymbol{\theta} = \tilde{\mathbf{h}}_{\boldsymbol{\theta}}^t$, by plugging in the expression for $\tilde{\mathbf{y}}^t$ from subsection 3.2, proposition 3.1 states that

$$\frac{\mathrm{d}\tilde{\mathbf{y}}^t}{\mathrm{d}\boldsymbol{\theta}} = \tilde{\mathbf{y}}_{\mathbf{h}^t}^t\left[(\prod_{i=1}^{t-1}\mathbf{h}_{\mathbf{h}^{t-i}}^{t-i+1})\mathbf{h}_{\boldsymbol{\theta}}^1 + (\prod_{i=1}^{t-2}\mathbf{h}_{\mathbf{h}^{t-i}}^{t-i+1})\mathbf{h}_{\boldsymbol{\theta}}^2 + \cdots + \mathbf{h}_{\mathbf{h}^{t-1}}^t\mathbf{h}_{\boldsymbol{\theta}}^{t-1} + \mathbf{h}_{\boldsymbol{\theta}}^t\right] + \tilde{\mathbf{y}}_{\boldsymbol{\theta}}^t.$$

In the context of SSM, we have:

$$\mathbf{h}^t = \mathbf{A}^t\mathbf{h}^{t-1} + \mathbf{B}^t\hat{\mathbf{x}}^t, \mathbf{h}_{\mathbf{h}^{t-1}}^t = \mathbf{A}^t, \mathbf{h}_{\boldsymbol{\theta}}^t = \mathbf{A}_{\boldsymbol{\theta}}^t\mathbf{h}^{t-1} + \mathbf{B}_{\boldsymbol{\theta}}^t\hat{\mathbf{x}}^t, \tilde{\mathbf{y}}^t = \mathbf{C}^t\mathbf{h}^t, \tilde{\mathbf{y}}_{\mathbf{h}^t}^t = \mathbf{C}^t, \tilde{\mathbf{y}}_{\boldsymbol{\theta}}^t = \mathbf{C}_{\boldsymbol{\theta}}^t\mathbf{h}^t. \tag{8}$$

Plugging in these relations, we get:

$$\frac{\mathrm{d}\tilde{\mathbf{y}}^t}{\mathrm{d}\boldsymbol{\theta}} = \mathbf{C}^t\left[(\prod_{i=1}^{t-1}\mathbf{A}^{t+1-i})\mathbf{h}_{\boldsymbol{\theta}}^1 + (\prod_{i=1}^{t-2}\mathbf{A}^{t+1-i})\mathbf{h}_{\boldsymbol{\theta}}^2 + \cdots + (\prod_{i=1}^{2}\mathbf{A}^{t+1-i})\mathbf{h}_{\boldsymbol{\theta}}^{t-2} + \mathbf{A}^t\mathbf{h}_{\boldsymbol{\theta}}^{t-1} + \mathbf{h}_{\boldsymbol{\theta}}^t\right] + \tilde{\mathbf{y}}_{\boldsymbol{\theta}}^t. \tag{9}$$

Define the adjoint state $\boldsymbol{\lambda}^{t,\tau} = \mathbf{C}^t(\prod_{i=1}^{t-\tau}\mathbf{A}^{t+1-i})$, we have

$$\frac{\mathrm{d}\tilde{\mathbf{y}}^t}{\mathrm{d}\boldsymbol{\theta}} = \boldsymbol{\lambda}^{t,1}\mathbf{h}_{\boldsymbol{\theta}}^1 + \boldsymbol{\lambda}^{t,2}\mathbf{h}_{\boldsymbol{\theta}}^2 + \cdots + \boldsymbol{\lambda}^{t,t-1}\mathbf{h}_{\boldsymbol{\theta}}^{t-1} + \boldsymbol{\lambda}^{t,t}\mathbf{h}_{\boldsymbol{\theta}}^t + \tilde{\mathbf{y}}_{\boldsymbol{\theta}}^t$$

Therefore, we have

$$\begin{aligned}
\frac{\mathrm{d}l^t}{\mathrm{d}\boldsymbol{\theta}} &= \frac{\mathrm{d}l^t}{\mathrm{d}\mathbf{y}^t}\frac{\mathrm{d}(\tilde{\mathbf{y}}^t + \hat{\mathbf{x}}^t)}{\mathrm{d}\boldsymbol{\theta}} \\
&= \frac{\mathrm{d}l^t}{\mathrm{d}\mathbf{y}^t}\frac{\mathrm{d}\tilde{\mathbf{y}}^t}{\mathrm{d}\boldsymbol{\theta}} \\
&= \frac{\mathrm{d}l^t}{\mathrm{d}\mathbf{y}^t}[\boldsymbol{\lambda}^{t,1}\mathbf{h}_{\boldsymbol{\theta}}^1 + \boldsymbol{\lambda}^{t,2}\mathbf{h}_{\boldsymbol{\theta}}^2 + \cdots + \boldsymbol{\lambda}^{t,t-1}\mathbf{h}_{\boldsymbol{\theta}}^{t-1} + \boldsymbol{\lambda}^{t,t}\mathbf{h}_{\boldsymbol{\theta}}^t + \tilde{\mathbf{y}}_{\boldsymbol{\theta}}^t]
\end{aligned}$$

Plug in everything, we have

$$\begin{aligned}
\frac{\mathrm{d}l^t}{\mathrm{d}\boldsymbol{\theta}} &= \frac{\mathrm{d}l^t}{\mathrm{d}\mathbf{y}^t}[\boldsymbol{\lambda}^{t,1}(\mathbf{A}_{\boldsymbol{\theta}}^1\mathbf{h}^0 + \mathbf{B}_{\boldsymbol{\theta}}^1\hat{\mathbf{x}}^1) + \boldsymbol{\lambda}^{t,2}(\mathbf{A}_{\boldsymbol{\theta}}^2\mathbf{h}^1 + \mathbf{B}_{\boldsymbol{\theta}}^2\hat{\mathbf{x}}^2) + \cdots + \boldsymbol{\lambda}^{t,t}(\mathbf{A}_{\boldsymbol{\theta}}^t\mathbf{h}^{t-1} + \mathbf{B}_{\boldsymbol{\theta}}^t\hat{\mathbf{x}}^t) + \mathbf{C}_{\boldsymbol{\theta}}^t\mathbf{h}^t \\
&= \left[\sum_{i=1}^t\frac{\mathrm{d}l^t}{\mathrm{d}\mathbf{y}^t}\boldsymbol{\lambda}^{t,i}(\mathbf{A}_{\boldsymbol{\theta}}^i\mathbf{h}^{i-1} + \mathbf{B}_{\boldsymbol{\theta}}^i\hat{\mathbf{x}}^i)\right] + \frac{\mathrm{d}l^t}{\mathrm{d}\mathbf{y}^t}\mathbf{C}_{\boldsymbol{\theta}}^t\mathbf{h}^t \\
&= \left[\sum_{i=1}^t\mathrm{vjp}_{\mathcal{A}^i}(\frac{\mathrm{d}l^t}{\mathrm{d}\mathbf{y}^t}\boldsymbol{\lambda}^{t,i}\otimes\mathbf{h}^{i-1}) + \mathrm{vjp}_{\mathcal{B}^i}(\frac{\mathrm{d}l^t}{\mathrm{d}\mathbf{y}^t}\boldsymbol{\lambda}^{t,i}\otimes\hat{\mathbf{x}}^i)\right] + \mathrm{vjp}_{\mathcal{C}^t}(\frac{\mathrm{d}l^t}{\mathrm{d}\mathbf{y}^t}\otimes\mathbf{h}^t)
\end{aligned}$$

where we define $\mathrm{vjp}_{NN^i}(v) = v \cdot NN_{\boldsymbol{\theta}}(\mathrm{Input}^i)$, with $\boldsymbol{\theta}$ being $NN$'s parameters and $i$ being the index of Input. Now, as $\mathrm{vjp}_{\boldsymbol{\mathcal{A}}^i}(\frac{\mathrm{d}l^t}{\mathrm{d}\mathbf{y}^t}\boldsymbol{\lambda}^{t,i} \otimes \mathbf{h}^{i-1})$, $\mathrm{vjp}_{\boldsymbol{\mathcal{B}}^i}(\frac{\mathrm{d}l^t}{\mathrm{d}\mathbf{y}^t}\boldsymbol{\lambda}^{t,i} \otimes \hat{\mathbf{x}}^i)$, and $\mathrm{vjp}_{\boldsymbol{\mathcal{C}}^t}(\frac{\mathrm{d}l^t}{\mathrm{d}\mathbf{y}^t} \otimes \mathbf{h}^t)$ are separate, we have

$$\frac{\mathrm{d}l^t}{\mathrm{d}\boldsymbol{\theta}} = \left[\sum_{i=1}^{t}\mathrm{vjp}_{\boldsymbol{\mathcal{A}}^i}(\frac{\mathrm{d}l^t}{\mathrm{d}\mathbf{y}^t}\boldsymbol{\lambda}^{t,i} \otimes \mathbf{h}^{i-1})\right] \oplus \left[\sum_{i=1}^{t}\mathrm{vjp}_{\boldsymbol{\mathcal{B}}^i}(\frac{\mathrm{d}l^t}{\mathrm{d}\mathbf{y}^t}\boldsymbol{\lambda}^{t,i} \otimes \hat{\mathbf{x}}^i)\right] \oplus \mathrm{vjp}_{\boldsymbol{\mathcal{C}}^t}(\frac{\mathrm{d}l^t}{\mathrm{d}\mathbf{y}^t} \otimes \mathbf{h}^t), \quad (10)$$

where $\oplus$ is vector concatenation. $\qquad\square$

## A.2 Proof for proposition 4.2

*Proof.* First, using the structure of ResNet, we have

$$\begin{aligned}
\frac{\mathrm{d}L}{\mathrm{d}\boldsymbol{\theta}} &= \sum_{t=1}^{T}\frac{\mathrm{d}l^t}{\mathrm{d}\mathbf{y}_K^t}\frac{\mathrm{d}\mathbf{y}_K^t}{\mathrm{d}\boldsymbol{\theta}} \\
&= \sum_{t=1}^{T}\frac{\mathrm{d}l^t}{\mathrm{d}\mathbf{y}_K^t}\frac{\mathrm{d}(\mathbf{y}_0^t + \sum_{k=1}^{K}\tilde{\mathbf{y}}_k^t)}{\mathrm{d}\boldsymbol{\theta}} \\
&= \sum_{t=1}^{T}\frac{\mathrm{d}l^t}{\mathrm{d}\mathbf{y}_K^t}\sum_{k=1}^{K}\frac{\mathrm{d}\tilde{\mathbf{y}}_k^t}{\mathrm{d}\boldsymbol{\theta}} \\
&= \sum_{t=1}^{T}\sum_{k=1}^{K}\frac{\mathrm{d}l^t}{\mathrm{d}\mathbf{y}_K^t}\frac{\mathrm{d}\tilde{\mathbf{y}}_k^t}{\mathrm{d}\boldsymbol{\theta}}
\end{aligned}$$

from proposition 4.1, we have proven that for a single SSM model, we have

$$\frac{\mathrm{d}l^t}{\mathrm{d}\boldsymbol{\theta}} = \left[\sum_{i=1}^{t}\mathrm{vjp}_{\boldsymbol{\mathcal{A}}^i}(\frac{\mathrm{d}l^t}{\mathrm{d}\mathbf{y}^t}\boldsymbol{\lambda}^{t,i} \otimes \mathbf{h}^{i-1})\right] \oplus \left[\sum_{i=1}^{t}\mathrm{vjp}_{\boldsymbol{\mathcal{B}}^i}(\frac{\mathrm{d}l^t}{\mathrm{d}\mathbf{y}^t}\boldsymbol{\lambda}^{t,i} \otimes \hat{\mathbf{x}}^i)\right] \oplus \mathrm{vjp}_{\boldsymbol{\mathcal{C}}^t}(\frac{\mathrm{d}l^t}{\mathrm{d}\mathbf{y}^t} \otimes \mathbf{h}^t),$$

so for the ResNet model, we have

$$\begin{aligned}
\frac{\mathrm{d}L}{\mathrm{d}\boldsymbol{\theta}} &= \sum_{t=1}^{T}\sum_{k=1}^{K}\frac{\mathrm{d}l^t}{\mathrm{d}\mathbf{y}_K^t}\frac{\mathrm{d}\tilde{\mathbf{y}}_k^t}{\mathrm{d}\boldsymbol{\theta}} \\
&= \sum_{t=1}^{T}\sum_{k=1}^{K}\left\{\left[\sum_{i=1}^{t}\mathrm{vjp}_{\boldsymbol{\mathcal{A}}_k^i}(\frac{\mathrm{d}l^t}{\mathrm{d}\mathbf{y}_K^t}\boldsymbol{\lambda}_k^{t,i} \otimes \mathbf{h}_k^{i-1})\right] \oplus \left[\sum_{i=1}^{t}\mathrm{vjp}_{\boldsymbol{\mathcal{B}}_k^i}(\frac{\mathrm{d}l^t}{\mathrm{d}\mathbf{y}_K^t}\boldsymbol{\lambda}_k^{t,i} \otimes \hat{\mathbf{x}}_k^i)\right] \oplus \mathrm{vjp}_{\boldsymbol{\mathcal{C}}_k^t}(\frac{\mathrm{d}l^t}{\mathrm{d}\mathbf{y}_K^t} \otimes \mathbf{h}_k^t)\right\} \\
&= \left(\sum_{t=1}^{T}\sum_{k=1}^{K}\mathrm{vjp}_{\boldsymbol{\mathcal{C}}_k^t}(\frac{\mathrm{d}l^t}{\mathrm{d}\mathbf{y}_K^t} \otimes \mathbf{h}_k^t)\right) \\
&\oplus \left(\sum_{t=1}^{T}\sum_{k=1}^{K}\sum_{i=1}^{t}\mathrm{vjp}_{\boldsymbol{\mathcal{A}}_k^i}(\frac{\mathrm{d}l^t}{\mathrm{d}\mathbf{y}_K^t}\boldsymbol{\lambda}_k^{t,i} \otimes \mathbf{h}_k^{i-1})\right) \\
&\oplus \left(\sum_{t=1}^{T}\sum_{k=1}^{K}\sum_{i=1}^{t}\mathrm{vjp}_{\boldsymbol{\mathcal{B}}_k^i}(\frac{\mathrm{d}l^t}{\mathrm{d}\mathbf{y}_K^t}\boldsymbol{\lambda}_k^{t,i} \otimes \hat{\mathbf{x}}_k^i)\right) \\
&= \left(\sum_{t=1}^{T}\sum_{k=1}^{K}\mathrm{vjp}_{\boldsymbol{\mathcal{C}}_k^t}(\frac{\mathrm{d}l^t}{\mathrm{d}\mathbf{y}_K^t} \otimes \mathbf{h}_k^t)\right) \\
&\oplus \left(\sum_{t=1}^{T}\sum_{k=1}^{K}\sum_{i=1}^{t}\mathrm{vjp}_{\boldsymbol{\mathcal{A}}_k^i}(\frac{\mathrm{d}l^t}{\mathrm{d}\mathbf{y}_K^t}\boldsymbol{\lambda}_k^{t,i} \otimes \mathbf{h}_k^{i-1})\right) \\
&\oplus \left(\sum_{t=1}^{T}\sum_{k=1}^{K}\sum_{i=1}^{t}\mathrm{vjp}_{\boldsymbol{\mathcal{B}}_k^i}(\frac{\mathrm{d}l^t}{\mathrm{d}\mathbf{y}_K^t}\boldsymbol{\lambda}_k^{t,i} \otimes \hat{\mathbf{y}}_{k-1}^i)\right)
\end{aligned} \quad (11)$$

where the input to $\mathrm{vjp}_{\mathcal{C}_k^t}(\frac{\mathrm{d}l^t}{\mathrm{d}\mathbf{y}_K^t} \otimes \mathbf{h}_k^t)$, $\mathrm{vjp}_{\mathcal{A}_k^i}(\frac{\mathrm{d}l^t}{\mathrm{d}\mathbf{y}_K^t}\boldsymbol{\lambda}_k^{t,i} \otimes \mathbf{h}_k^{i-1})$, and $\mathrm{vjp}_{\mathcal{B}_k^i}(\frac{\mathrm{d}l^t}{\mathrm{d}\mathbf{y}_K^t}\boldsymbol{\lambda}_k^{t,i} \otimes \hat{\mathbf{y}}_{k-1}^i)$ are computed with the k-th SSM and the $\hat{\mathbf{x}}_k^i = \hat{\mathbf{y}}_{k-1}^i = \mathrm{RMSNorm}(\mathbf{y}_{k-2}^i + \mathrm{SSM}_{k-1}(\hat{\mathbf{Y}}_{k-2})^i)$ (the normalized output sequence of the (k-1)-th SSM), and the adjoint state $\boldsymbol{\lambda}_k^{t,\tau} = \mathbf{C}_k^t(\prod_{i=1}^{t-\tau} \mathbf{A}_k^{t+1-i})$.

$\square$

### A.3 Proof of concept for VJP computation

As a proof of concept of why $(\mathrm{d}l^t/\mathrm{d}\mathbf{y}^t)\mathbf{C}_{\boldsymbol{\theta}}^t\mathbf{h}^t$ can computed with vjp, we present an explicit and simple example. We have $\mathbf{y} = [y_1, y_2]$, $\mathbf{h} = [h_1, h_2, h_3]$, $\boldsymbol{\theta} = \vec{\boldsymbol{\theta}}$. We then have

$$\frac{dl}{d\mathbf{y}} = \begin{bmatrix} l_{y_1} & l_{y_2} \end{bmatrix} \in \mathbb{R}^{1 \times P}$$

$$\mathbf{C}_{\boldsymbol{\theta}} = \begin{bmatrix} C_{11}^{\vec{\boldsymbol{\theta}}} & C_{12}^{\vec{\boldsymbol{\theta}}} & C_{13}^{\vec{\boldsymbol{\theta}}} \\ C_{21}^{\vec{\boldsymbol{\theta}}} & C_{22}^{\vec{\boldsymbol{\theta}}} & C_{23}^{\vec{\boldsymbol{\theta}}} \end{bmatrix} \in \mathbb{R}^{P \times N \times |\boldsymbol{\theta}|}$$

$$\mathbf{h} = \begin{bmatrix} h_1 \\ h_2 \\ h_3 \end{bmatrix} \in \mathbb{R}^{N \times 1}$$

With each $C_{ij}^{\vec{\boldsymbol{\theta}}} = [\partial C_{ij}/\partial \boldsymbol{\theta}_1, \ldots, \partial C_{ij}/\partial \boldsymbol{\theta}_{|\boldsymbol{\theta}|}] \in \mathbb{R}^{|\boldsymbol{\theta}|}$. We have

$$\frac{\mathrm{d}l}{\mathrm{d}y}\mathbf{C}_{\boldsymbol{\theta}}\mathbf{h} = C_{11}^{\vec{\boldsymbol{\theta}}}l_{y_1}h_1 + C_{21}^{\vec{\boldsymbol{\theta}}}l_{y_2}h_1 + C_{12}^{\vec{\boldsymbol{\theta}}}l_{y_1}h_2 + C_{22}^{\vec{\boldsymbol{\theta}}}l_{y_2}h_2 + C_{13}^{\vec{\boldsymbol{\theta}}}l_{y_1}h_3 + C_{23}^{\vec{\boldsymbol{\theta}}}l_{y_2}h_3$$

$$= [l_{y_1}h_1 \; l_{y_1}h_2 \; l_{y_1}h_3 \; l_{y_2}h_1 \; l_{y_2}h_2 \; l_{y_2}h_3] \cdot [C_{11}^{\vec{\boldsymbol{\theta}}} \; C_{12}^{\vec{\boldsymbol{\theta}}} \; C_{13}^{\vec{\boldsymbol{\theta}}} C_{21}^{\vec{\boldsymbol{\theta}}} \; C_{22}^{\vec{\boldsymbol{\theta}}} \; C_{23}^{\vec{\boldsymbol{\theta}}}]$$

$$= \mathrm{sum}\left( \left( \begin{bmatrix} l_{y_1} \\ l_{y_2} \end{bmatrix} \otimes \begin{bmatrix} h_1 & h_2 & h_3 \end{bmatrix} \right) \circ \begin{bmatrix} C_{11}^{\vec{\boldsymbol{\theta}}} & C_{12}^{\vec{\boldsymbol{\theta}}} & C_{13}^{\vec{\boldsymbol{\theta}}} \\ C_{21}^{\vec{\boldsymbol{\theta}}} & C_{22}^{\vec{\boldsymbol{\theta}}} & C_{23}^{\vec{\boldsymbol{\theta}}} \end{bmatrix} \right)$$

where $\cdot$ is vector dot product, $\otimes$ is vector outer product, $\circ$ is element-wise product, and sum means summing all elements in a matrix.

## B Distributed tensors' locations

We provide the specific location for each tensors in distributed training:

Table 2: Tensors stored on each GPU, part 1.

| GPU index | $\mathrm{d}l(\mathbf{o}^t)/\mathrm{d}y_K^t$ | $h_k^t$ |
|---|---|---|
| $\upsilon = 1$ | $t = 1, \ldots, T$ | $t = 1, \ldots, T; \; k = 1, \ldots K//\Upsilon$ |
| $\upsilon = 2$ | $t = 1, \ldots, T$ | $t = 1, \ldots, T; \; k = K//\Upsilon + 1, \ldots, 2(K//\Upsilon)$ |
| $\ldots$ | $\ldots$ | $\ldots$ |
| $\upsilon = \Upsilon - 1$ | $t = 1, \ldots, T$ | $t = 1, \ldots, T; \; k = (\Upsilon - 2)(K//\Upsilon) + 1, \ldots, (\Upsilon - 1)(K//\Upsilon)$ |
| $\upsilon = \Upsilon$ | $t = 1, \ldots, T$ | $t = 1, \ldots, T; \; k = (\Upsilon - 1)(K//\Upsilon) + 1, \ldots, K$ |

Table 3: Tensors stored on each GPU, part 2.

| GPU index | $C_k^t$ |
|---|---|
| $\upsilon = 1$ | $t = 1, \ldots, T; \ k = 1, \ldots K//\Upsilon$ |
| $\upsilon = 2$ | $t = 1, \ldots, T; \ k = K//\Upsilon + 1, \ldots, 2(K//\Upsilon)$ |
| $\ldots$ | $\ldots$ |
| $\upsilon = \Upsilon - 1$ | $t = 1, \ldots, T$ |
| $\upsilon = \Upsilon$ | $t = 1, \ldots, T; \ k = (\Upsilon - 1)(K//\Upsilon) + 1, \ldots, K$ |

Table 4: Tensors stored on each GPU, part 3.

| GPU index | $\hat{y}_k^t$ |
|---|---|
| $\upsilon = 1$ | $t = 1, \ldots, T; \ k = 0, \ldots K//\Upsilon - 1$ |
| $\upsilon = 2$ | $t = 1, \ldots, T; \ k = K//\Upsilon, \ldots, 2(K//\Upsilon) - 1$ |
| $\ldots$ | $\ldots$ |
| $\upsilon = \Upsilon - 1$ | $t = 1, \ldots, T; \ k = (\Upsilon - 2)(K//\Upsilon), \ldots, (\Upsilon - 1)(K//\Upsilon) - 1$ |
| $\upsilon = \Upsilon$ | $t = 1, \ldots, T; \ k = (\Upsilon - 1)(K//\Upsilon), \ldots, K - 1$ |

Table 5: Tensors stored on each GPU, part 4.

| GPU index | $A_k^t$ |
|---|---|
| $\upsilon = 1$ | $t = 2, \ldots, T; \ k = 1, \ldots K//\Upsilon$ |
| $\upsilon = 2$ | $t = 2, \ldots, T; \ k = K//\Upsilon + 1, \ldots, 2(K//\Upsilon)$ |
| $\ldots$ | $\ldots$ |
| $\upsilon = \Upsilon - 1$ | $t = 2, \ldots, T; \ k = (\Upsilon - 2)(K//\Upsilon) + 1, \ldots, (\Upsilon - 1)(K//\Upsilon)$ |
| $\upsilon = \Upsilon$ | $t = 2, \ldots, T; \ k = (\Upsilon - 1)(K//\Upsilon) + 1, \ldots, K$ |

Table 6: Tensors stored on each GPU, part 5.

| GPU index | $\boldsymbol{\theta}_k$ | Gradient$_k$ |
|---|---|---|
| $\upsilon = 1$ | $k = 1, \ldots K//\Upsilon$ | $k = 1, \ldots K//\Upsilon$ |
| $\upsilon = 2$ | $k = K//\Upsilon + 1, \ldots, 2(K//\Upsilon)$ | $k = K//\Upsilon + 1, \ldots, 2(K//\Upsilon)$ |
| $\ldots$ | $\ldots$ | $\ldots$ |
| $\upsilon = \Upsilon - 1$ | $k = (\Upsilon - 2)(K//\Upsilon) + 1, \ldots, (\Upsilon - 1)(K//\Upsilon)$ | $k = (\Upsilon - 2)(K//\Upsilon) + 1, \ldots, (\Upsilon - 1)(K//\Upsilon)$ |
| $\upsilon = \Upsilon$ | $k = (\Upsilon - 1)(K//\Upsilon) + 1, \ldots, K$ | $k = (\Upsilon - 1)(K//\Upsilon) + 1, \ldots, K$ |

