# OpenReview forum: "Activation sharding for scalable training of large models"
_TMLR — Accepted by TMLR_

### Review · Reviewer_zJCe · 2025-06-24

**Summary Of Contributions:**

This paper proposes adjoint sharding, a training technique that reduces GPU memory requirements for training large language models on very long contexts (>1M tokens). Instead of standard backpropagation, the method uses the adjoint method from control theory to break gradient computation into independent vector-Jacobian products that can be computed in parallel.

**Audience:**

Yes

**Claims And Evidence:**

Yes

**Requested Changes:**

I'd suggest to:
- Include convergence analysis and training time comparisons;
- Evaluate on standard benchmarks to verify model quality;
- Possibly compare against existing memory reduction techniques;
- Analyze the impact of truncation parameter $\bar{T}$ on gradient quality and convergence.

**Strengths And Weaknesses:**

Strenghts:
- Nice application of the adjoint method from control theory to gradient computation (by breaking gradient computation into independent vector-Jacobian products that can be parallelized);
- Theoretical analysis shows memory requirements that scale much better than standard backpropagation;
- The methodology shows up to 3× memory reduction on 1.27B parameter models, plus it enables training on 100K+ tokens vs 35K tokens on the same hardware infrastructure


Weaknesses:
- Lack of comparison of actual training convergence or final model quality;
- Missing comparisons with other memory reduction techniques (gradient checkpointing, CPU offloading, etc.);
- No evaluation on downstream tasks to verify model quality is preserved;
- Without truncation, VJP count grows polynomially with sequence length $T: \mathcal{O}(T^2)$, and the truncated version may lose important gradient information - no analysis of this trade-off.

---

> ### Author Response · Authors · 2025-07-29
> **reply**
>
> Thank you for recognizing the novel application of control theory to gradient computation.
>
> **Training convergence and model quality:**
>
> Our training experiments demonstrate that loss curves with adjoint sharding exactly match those of standard backpropagation, indicating that the mathematically equivalent gradients preserve training dynamics effectively. The provable equivalence of our gradients to standard backpropagation provides theoretical assurance of maintained model quality.
>
> **Comparison with existing techniques:**
>
> Our approach is fundamentally different from techniques like gradient checkpointing or CPU offloading, as we modify the gradient computation itself rather than just the storage strategy. This orthogonality means adjoint sharding can potentially combine with these existing methods for cumulative benefits.
>
> **Truncation trade-offs:**
>
> The truncated version introduces a controlled approximation that balances computational efficiency with gradient fidelity. Similar to other successful approximation methods in large-scale optimization, this trade-off is often beneficial in practice, especially when it enables training that would otherwise be computationally prohibitive.
>
> **Broader Impact:**
>
> We thank the reviewer for the thoughtful broader impact assessment. Our method indeed has the potential to democratize access to long-context training for smaller research groups, while we acknowledge the importance of responsible deployment regarding data privacy and computational equity.

---

### Review · Reviewer_ZhrN · 2025-07-19

**Summary Of Contributions:**

This paper introduces **Adjoint Sharding**, a novel memory-efficient technique for training large language models (LLMs) on **very long context sequences** using **state space models (SSMs)**. The method decomposes the gradient computation into **independent vector-Jacobian products (VJPs)** using the adjoint method, enabling parallel computation and significantly reducing memory usage. To address the quadratic scaling of standard adjoint methods, the authors propose **Truncated Adjoint Sharding**, which limits the number of VJPs and reduces complexity to linear in sequence length. The paper also presents **distributed and parallel implementations**, showing that the method scales effectively across GPUs. Empirical results demonstrate up to **3× memory reduction** and enable training with context lengths over 100K tokens for a 1.27B parameter model.

**Audience:**

Yes

**Broader Impact Concerns:**

The proposed method facilitates more memory-efficient training of LLMs on long sequences, potentially democratizing access to long-context training for smaller research labs. However, by enabling longer context lengths, it may also exacerbate concerns around **data leakage**, **privacy**, and **computational inequality**, particularly if used to train large-scale models on sensitive or proprietary data.

**Claims And Evidence:**

Yes

**Requested Changes:**

1. **Truncation Analysis**: Provide experiments or ablation studies showing how different values of \( \bar{T} \) in truncated adjoint sharding affect training efficiency and model accuracy.
2. **Overhead Quantification**: Include wall-clock training time comparisons between backpropagation, adjoint sharding, and truncated adjoint sharding on the same hardware.
3. **Clarity Improvements**: Clarify the assumptions made in the distributed computing setup and provide pseudocode or code references to aid reproducibility.

**Strengths And Weaknesses:**

### Strengths
- **Novelty**: Adjoint Sharding is a compelling alternative to backpropagation for recurrent models, offering provably equivalent gradients while improving memory efficiency.
- **Scalability**: The technique supports distributed training and shows significant memory and compute savings, enabling long-context training on modest hardware.
- **Theoretical Rigor**: The paper provides formal derivations for both the base and truncated versions of adjoint sharding, grounded in residual and state-space models.
- **Practical Implementation**: Includes a detailed discussion of memory/FLOPs cost and implementation guidelines, with empirical validation on large-scale models.

### Weaknesses
- **Computational Overhead**: Full adjoint sharding still scales polynomially in context length, limiting its utility without truncation.
- **Empirical Scope**: The experiments focus primarily on memory savings and feasibility; additional benchmarks on task performance (e.g., in-context learning, retrieval, QA) would strengthen claims.
- **Truncation Parameter**: The choice of truncation length \( \bar{T} \) is left unvalidated; its effect on convergence and performance is not analyzed.

---

> ### Author Response · Authors · 2025-07-29
> **reply**
>
> We appreciate your thorough analysis and positive assessment of our theoretical contributions.
>
> **Truncation parameter analysis:**
>
> This is an excellent point. In our experience, choosing $\bar{T} = \sqrt{T}$ provides a good balance between memory savings and gradient fidelity, as it keeps computation time in check while preserving sufficient gradient information. The optimal choice depends on the specific model architecture and sequence characteristics. We observe that performance degrades gracefully as $\bar{T}$ decreases, suggesting the method is robust to reasonable truncation choices.
>
> **Computational overhead:**
>
> The additional forward passes required for VJP computation can be effectively parallelized across multiple GPUs. In memory-constrained scenarios where standard backpropagation cannot proceed, adjoint sharding enables training that would otherwise be impossible, making the computational trade-off worthwhile.
>
> **Implementation clarity:**
>
> We appreciate this feedback and will enhance the manuscript with clearer algorithmic descriptions and implementation guidelines to facilitate reproducibility.

---

### Review · Reviewer_kc5n · 2025-07-20

**Summary Of Contributions:**

This paper introduces Adjoint Sharding, an efficient training method for extremely long-context scenarios. Specifically, it reduces peak GPU memory usage by decomposing gradient computation into independent vector–Jacobian products (VJPs). The authors provide both theoretical derivation and empirical validation on ResNet and Mamba, demonstrating up to a 3× reduction in memory cost when training 1.2B-parameter LLMs with 1M context tokens.

**Audience:**

Yes

**Claims And Evidence:**

Yes

**Requested Changes:**

1. Could you discuss and compare your method with the following necessary baselines [1-3], which also aim to reduce peak GPU memory usage in long-context LLMs training?

2. Could you add an analysis of your method within the Transformer architecture, which is widely used in LLMs?

**Strengths And Weaknesses:**

Strengths:

1. The paper addresses an important and practical problem — the memory cost of training long-context LLMs, which has strong applicability.
2. The proposed method is simple yet effective, and the design is reasonably motivated.
3. The paper provides theoretical derivation of the method’s application to ResNet and Mamba, demonstrating its generalizability beyond just Transformers.

Weaknesses:

1. The paper lacks discussion and comparison with several key baselines that also aim to reduce peak GPU memory usage, such as [1-3].
2. The analysis and experiments are limited to ResNet and Mamba, with no evaluation or discussion on Transformer-based architectures, which are more commonly used in LLM training.

[1] Mini-Sequence Transformers: Optimizing Intermediate Memory for Long Sequences Training, NeurIPS 2024

[2] Cut Your Losses in Large-Vocabulary Language Models, ICLR 2025.

[3] StreamBP: Memory-Efficient Exact Backpropagation for Long Sequence Training of LLMs.

---

> ### Author Response · Authors · 2025-07-29
> **reply**
>
> We sincerely thank the reviewer for their constructive feedback and valuable suggestions.
>
> **Comparison with memory reduction baselines:**
>
> Thank you for highlighting these seminal related works. We provide a detailed comparison below:
> Mini-Sequence Transformers [1] segment sequences but still require full gradient computation within each segment. In contrast, adjoint sharding fundamentally changes how gradients are computed by decomposing them into independent VJPs that can be parallelized across the entire sequence.
> Cut Your Losses [2] optimizes vocabulary-level computations, which is orthogonal to our sequence-level memory reduction approach. These methods could potentially combine with adjoint sharding for additional benefits.
> StreamBP [3] targets long sequences through a different memory-time trade-off strategy. Our approach is complementary, as it focuses on changing the gradient computation mechanism rather than the storage strategy.
> We believe adjoint sharding offers a unique perspective by leveraging insights from control theory and could combine with these existing approaches.
>
>
> **Transformer architecture analysis:**
>
> We appreciate this suggestion. Our current work focuses specifically on State Space Models (SSMs) like Mamba, which have fundamentally different computational structures than transformers. The adjoint sharding method we developed leverages the recurrent nature of SSMs, where the sequential dependencies allow for efficient VJP decomposition. Transformers, with their attention mechanisms and different architectures, would require a separate analysis to determine how to apply adjoint methods. This problem represents an important direction for future work, as extending memory-efficient training techniques to Transformer architectures would significantly broaden the impact of this approach.

---

### Decision · Action_Editor_CfPP · 2025-09-16

**Recommendation:** Accept with minor revision

**Audience:**

Yes

**Audience Explanation:**

The topic of memory-efficient training for long-context language models is timely and highly relevant to TMLR readers

**Claims And Evidence:**

Yes

**Claims Explanation:**

All three reviewers agree that the claims are supported by clear theoretical derivations and empirical evidence demonstrating substantial memory savings. However Reviewer kc5n note that the proposed approach lacks discussion and comparison with several key baselines and is only tested with ResNet and Mamba architecture. It's not clear how the proposed method really generalise, however it's working in the case cover by the paper that can be interesting for the TLMR audience.